# Using large language models to accelerate communication for eye gaze typing users with ALS

Shanqing Cai [1] ✉, Subhashini Venugopalan [1], Katie Seaver[1], Xiang Xiao[1], Katrin Tomanek[1], Sri Jalasutram[1], Meredith Ringel Morris[1], Shaun Kane[1], Ajit Narayanan[1], Robert L. MacDonald[1], Emily Kornman[2], Daniel Vance[2], Blair Casey[2], Steve M. Gleason[2], Philip Q. Nelson[1] & Michael P. Brenner [1,3]

Accelerating text input in augmentative and alternative communication (AAC) is a long-standing area of research with bearings on the quality of life in individuals with profound motor impairments. Recent advances in large language models (LLMs) pose opportunities for re-thinking strategies for enhanced text entry in AAC. In this paper, we present SpeakFaster, consisting of an LLM-powered user interface for text entry in a highly-abbreviated form, saving 57% more motor actions than traditional predictive keyboards in offline simulation. A pilot study on a mobile device with 19 non-AAC participants demonstrated motor savings in line with simulation and relatively small changes in typing speed. Lab and field testing on two eye-gaze AAC users with amyotrophic lateral sclerosis demonstrated text-entry rates 29–60% above baselines, due to significant saving of expensive keystrokes based on LLM predictions. These findings form a foundation for further exploration of LLM-assisted text entry in AAC and other user interfaces.

The loss of efficient means of communication presents one of the greatest challenges to people living with amyotrophic lateral sclerosis (ALS). As one of the effects of ALS is to impair one's ability to move their body, thereby limiting typing or speaking[1], eye-gaze tracking is often the only remaining human-computer interface. Coupled with an on-screen keyboard, gaze tracking allows users to enter text for speech output and electronic messaging, providing a means for augmentative and alternative communication (AAC)[2]. However, gaze typing is slow (usually below 10 words per minute (WPM)[3–5], creating a gap of more than an order of magnitude below typical speaking rates (up to 190 WPM in English[6,7]). Simulated models of eye gaze typing suggest that even the theoretical maximum performance is significantly lower than typical speech[8], which suggests that unassisted, QWERTY-based eye typing will remain significantly slower than typical speech. This impairment to communication ability can have significant negative impact on one's ability to participate in social life[1,9,10].

With recent developments in Brain-Computer Interfaces (BCI), typing via brain signals becomes possible and is allowing disabled users to type at increasingly higher rates through imagined speech or hand movements[11,12], however these methods require invasively-implanted electrodes in the cerebral cortex and hence are not as widely tested or adopted as eye-gaze typing.

A major bottleneck to faster gaze typing for disabled users is the eye fatigue and temporal cost associated with performing many keystrokes[13]. This bottleneck applies to BCI text entry systems that operate at the level of individual characters[11,14] as well. One way to attack this bottleneck is to develop techniques to significantly reduce the number of keystrokes needed to enter text by predicting upcoming text from the preceding text and non-linguistic contextual signals[6,15–20]. For example, completion for partially-typed words and prediction of the next word could reduce the average number of keystrokes needed per word. However current keystroke-saving

[1]Google, Mountain View, CA, USA. [2]Team Gleason Foundation, New Orleans, LA, USA. [3]School of Engineering and Applied Sciences, Harvard University, Cambridge, MA, USA. ✉e-mail: cais@google.com

techniques used in AAC and commercial text-input software (e.g. SwiftKey[21], Gboard[22] and Smart Compose[23]) are still insufficient for closing the communication gaps for impaired users, as the spans of correct predictions are short, the utilisation of the predictions too infrequent and the awareness of context too limited to offset the cost of scanning predictions[13,24]. While AAC users may conserve effort by planning ahead and typing out text they wish to speak in future conversations, this tactic sacrifices spontaneity, which is an important component of interpersonal communication[9].

In the past few years, there has been a revolution in the development of large language models (LLMs)[25–28]. LLMs are initially trained on simple objectives such as predicting the next text token over a very large text corpus, such as public web text and book content, but then can be nudged to have emergent capabilities that approach or exceed human-level performance on language-based tasks ranging from answering difficult mathematics problems[29] to carrying on openended conversations[25]. These LLMs can make predictions with a much longer context than traditional n-gram language models (LMs) and thus offer the possibility of substantially accelerating character-based typing.

In this study, we report a text-entry user interface (UI) called SpeakFaster that leverages fine-tuned LLMs and conversational context to expand highly-abbreviated English text[30,31] (word initials, supplemented by additional letters and words when necessary) into the desired full phrases at very high accuracy. To demonstrate the efficacy of this new system, we first used simulations to show that the accuracy of the SpeakFaster method is sufficiently high to significantly accelerate character-based typing. Second, we integrated the SpeakFaster UI with the eye-gaze keyboards of two users with ALS and demonstrated a significant enhancement in text-entry speed (29–60%) and saving of motor actions compared to a baseline of conventional forward suggestions. On non-AAC participants, the overall speed of text entry was similar to conventional smart keyboards, while the motor

action savings was still substantial. Our findings also indicate that users were able to learn to use the system with relatively little practice (about 20–50 phrases).

## Results

### The SpeakFaster UI

Word completion and next-word prediction based on n-gram LMs[24,32] exploit the statistical dependence of a word on a small number of (typically up to four) preceding words. By contrast, LLMs are able to take advantage of broader context, including tens or hundreds of preceding words entered by the user and previous turns of the ongoing conversation. We have previously demonstrated[33] that a finetuned 64-billion-parameter Google LaMDA[25] model can expand abbreviations of the word-initial form (e.g. 'ishpitb') into full phrases (e.g. 'I saw him play in the bedroom', Fig. 1) at a top-5 exact-match accuracy of 48–77% when provided with conversational context, i.e. previous dialogue turn(s). Failures to find exact matches tend to occur on longer and more complex phrases. While promising, a practical solution needs to ensure that the user is able to type any arbitrary phrase in subsequent attempts in case of a failure in the initial abbreviation expansion (AE), i.e. the user will never run into a 'dead end' in the UI.

We therefore developed a UI and two underlying fine-tuned LLMs as a complete, practical solution. LLM 'KeywordAE' is capable of expanding abbreviations that mix initials with words that are fully or incompletely spelled out (Fig. 2). The KeywordAE model is also capable of expanding initials-only abbreviations, and hence provides a superset of the capabilities of the fine-tuned LLM in ref. 33 LLM 'FillMask' is capable of providing alternative words that begin with a given initial letter in the context of surrounding words (Fig. 3). The two models were each fine-tuned with ~1.8 million unique triplets of {*context*, *abbreviation*, *fullphrase*} synthesised from four public datasets of dialogues in English. Here, 'context' refers to the previous turns of the

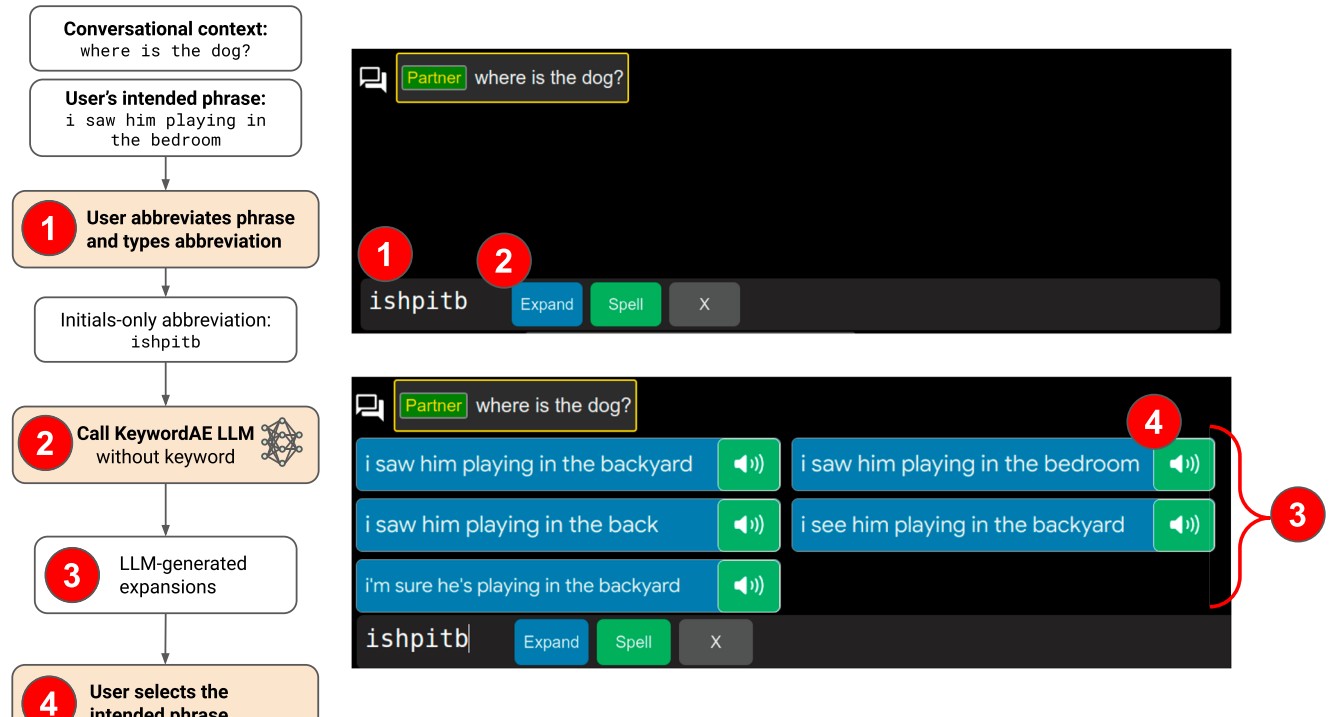

**Fig. 1 | The primary interaction pathway of abbreviated text entry in the SpeakFaster UI: the initials-only pathway.** The KeywordAE LLM serves this UI pathway This UI pathway requires the user to enter only the word-initial letters of the intended phrase. The 'Speaker' button to the right of each candidate phrase

(e.g. Label 4) allows the user to select and speak the correct phrase out via text-tospeech. Gaze-clicking the 'Expand' button (Label 2) is optional since calls to the LLM are triggered automatically following the eye-gaze keystrokes. The gaze-driven onscreen keyboard is omitted from the screenshots in this figure (also Figs. 2 and 3).

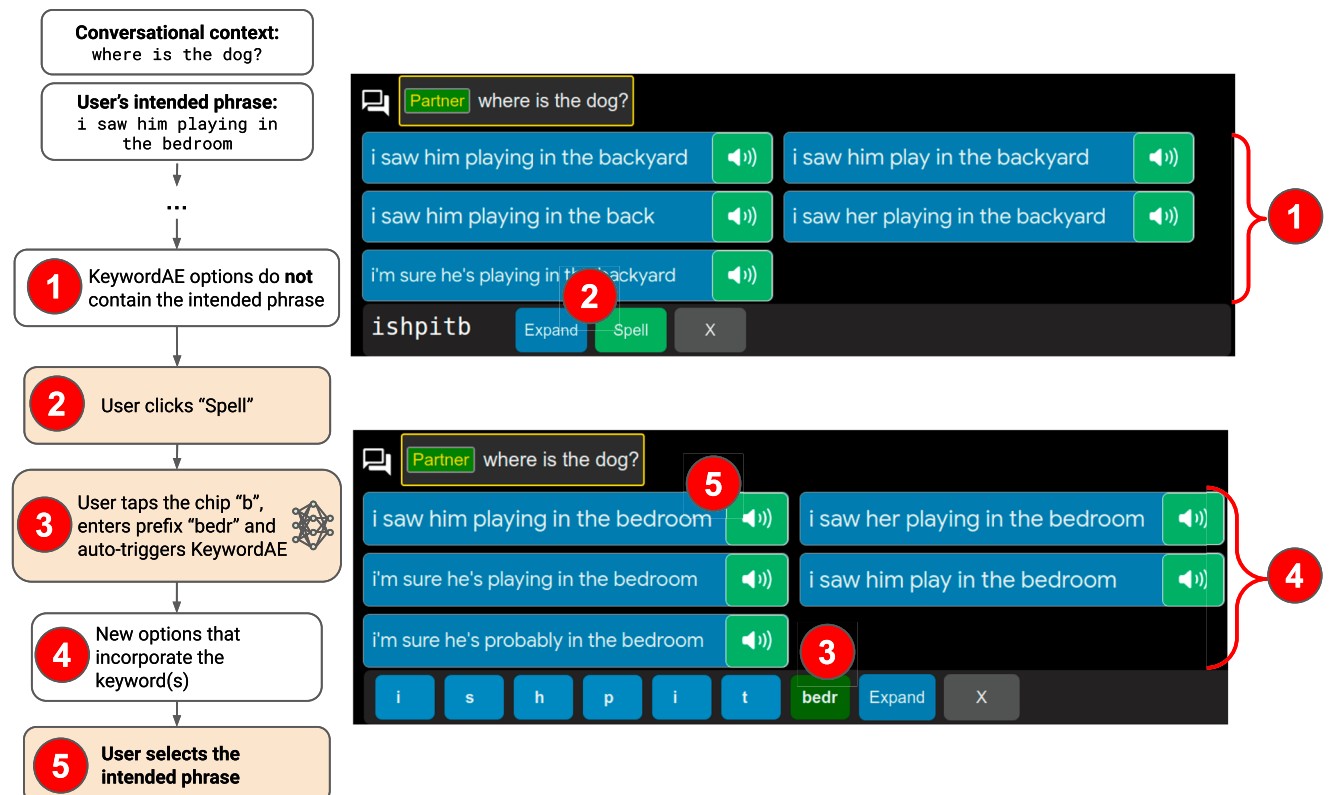

**Fig. 2 | The Keyword Abbreviation Expansion (KeywordAE) UI pathway.** This is an extension of initials-only AE (Fig. 1) that allows words spelled out fully or partially to be mixed with initials in the original abbreviation, in an order that matches to the intended phrase. We refer to such fully- or partially-spelled words as 'keywords'. They guide the system towards the intended phrase. Label 3 ('bedr' for 'bedroom') illustrates the support for partially-spelled keywords supported by KeywordAE v2, which differs from KeywordAE v1, where only fully-spelled keywords (e.g. 'bedroom') are supported.

ongoing dialogue, including the turns authored by the user and the conversation partner (See Supplementary Section 1.1 for details of LLM fine-tuning and evaluation).

To form conduits to the fine-tuned LLMs, we designed a UI with three pathways, namely initials-only AE, KeywordAE and FillMask, to support a complete abbreviated text input experience (see Fig. 1). The initials-only pathway is the common starting point of all phrase-entry workflows in the SpeakFaster UI. Among the three pathways, it involves the fewest keystrokes and gaze clicks and alone suffices for short and predictable phrases. The user starts by typing an initialism that represents the intended phrase (Label 1 in Fig. 1). The abbreviation is typed with a conventional soft keyboard or gaze-driven on-screen keyboard (e.g. Tobii ® Computer Control). As the user enters the abbreviation, the UI automatically triggers calls to the KeywordAE LLM after every keystroke, including the user-typed abbreviation along with all previous turns of the conversation as input to the LLM. Each call returns the top-5 most likely options based on the conversational context and the abbreviation, which are rendered in the UI for the user to peruse and select (Label 3 in Fig. 1). The number of options provided (5) is based on the screen size of gaze tablet devices and the point of 'diminishing return' for motor saving rate from offline simulation results (see next section). If one of the candidate phrases matches the intended phrase, the user selects the phrase by clicking the 'Speaker' button associated with it (Label 4 in Fig. 1), which dispatches the phrase for text-to-speech output and ends the phrase entry workflow. The selected phrase ('I saw him playing in the bedroom' in this example) becomes a part of the conversational context for future turns.

If the intended phrase is not found via the initials-only pathway, however, two alternative UI pathways are available to assist the user in finding the intended phrase. One of the pathways is KeywordAE. The user gaze-clicks the 'Spell' button (Label 2 in Fig. 2), which turns the abbreviation in the input bar into gaze-clickable chips, one for each character of the initials-only abbreviation (e.g. bottom of Fig. 2). The user selects a word to spell by gaze-clicking the corresponding chip. This turns the chip into an input box, in which the user types the word by using the on-screen keyboard (Label 3 in Fig. 2). Subsequent calls to the LLM will contain the partially- or fully-spelled words in addition to the initials of the unspelled words. A call to the KeywordAE is triggered automatically after every keystroke. After each call, the UI renders the latest top-5 phrase expansion returned by the KeywordAE LLM (Label 4 in Fig. 2). If the intended phrase is found, the user selects it by a gaze click of the speaker button as described before (Label 5 in Fig. 2). We constructed two versions of KeywordAE models: KeywordAE v1 requires each keyword to be typed in full, while KeywordAE v2 allows a keyword to be typed incompletely ('bedroom' as 'bedr'). Simulation results below show v2 leads to greater keystroke saving than v1.

The KeywordAE UI pathway is not limited to spelling out a single word. The UI allows the user to spell multiple words, which is necessary for longer and more unpredictable phrases. In the unlikely case where the AE LLM predicts none of the words of the sentence correctly, the KeywordAE pathway reduces to spelling out all the words of the phrase.

FillMask is another way to recover from the failure to find the exact intended phrase. Unlike KeywordAE, FillMask only suits the cases in which very few words (typically one word) of an expansion are incorrect (i.e. the phrase is a 'near miss'). For instance, one of the phrases 'I saw him play in the backyard' missed the intended phrase 'I saw him play in the bedroom' by only one incorrect word ('backyard', Label 2 in Fig. 3). The user clicks the near-miss phrase, which causes the words of the phrase to appear as chips in the input bar. Clicking the chip that corresponds to the incorrect word ('backyard', Label 3 in Fig. 3) triggers a call to the FillMask LLM, the response from which

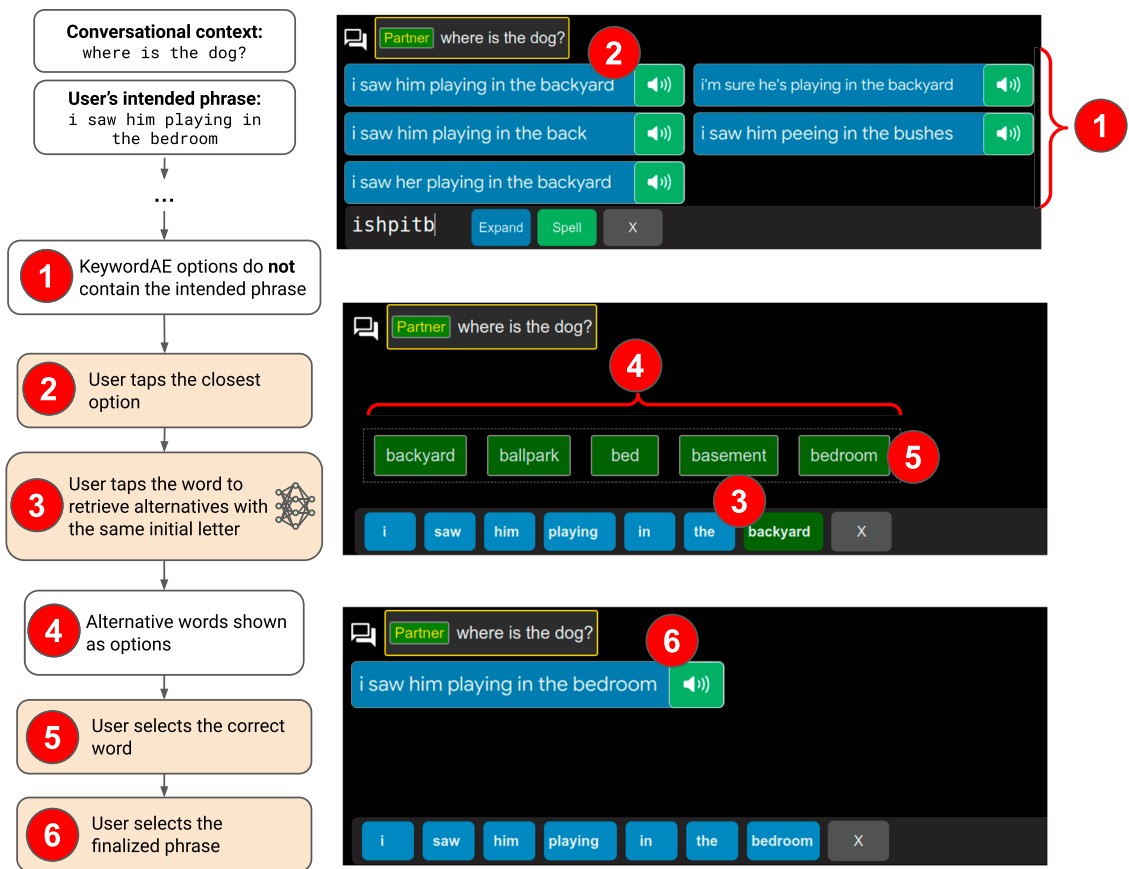

**Fig. 3 | The FillMask UI pathway.** This is an additional interaction flow that allows users to recover from failure to find the full phrase from the initials-only abbreviation. The FillMask UI allows the user to find sensible replacements for an incorrect word that starts with the same letter.

contains alternative words that start with the same initial letter and fit the context formed by the other words of the sentence and by the previous turn(s) of the conversation. The user selects the correct word (Label 5) by clicking it and then clicking the speaker button to finalise the phrase entry (Label 6).

In addition to the single-word interaction shown in Fig. 3, the FillMask pathway allows the UI to replace multiple words (or do replacement multiple times in a given word slot) after the initial replacement. In rare cases where the FillMask LLM fails to provide the intended word, the user can fall back to typing the correct word in the input box by using the eye-gaze keyboard.

As shown above, KeywordAE and FillMask are two alternative interaction modes for recovering from a failure to obtain the intended phrase via the initials-only pathway. The user's decision on which pathway to choose should be determined by whether a near-miss option exists. This proposition is supported by simulation results in the next section. In the current study, the SpeakFaster UI allows the user to use the FillMask mode after using the KeywordAE mode, which is useful for finding the correct words in hard-to-predict phrases. But entering the KeywordAE mode is not allowed after using the FillMask mode, because FillMask should be used only during the final phase of a phrase entry workflow, where all but one or two words of a candidate phrase are correct. These heuristics and design considerations of the UI pathways were made clear to the users through initial training and practice at the beginning of the user studies described below. The SpeakFaster UI is only one of many possible UI designs for supporting AAC text entry with LLMs[19,20]. Its justification comes from prior studies on LLM's capabilities in expanding abbreviations[33], its consistency with the conventional lookup-based AE in AAC[30] and the empirical results from the user studies reported below.

## Simulation results

To measure the approximate upper bound of the motor-action savings in this our text-entry UI, we ran simulation on the test split of a corrected version of the Turk Dialogues corpus (TDC)[33,34]. To simulate an ideal user's actions in the SpeakFaster UI when entering the text for a dialogue turn, we first invoked AE without keywords (Fig. 1). If the matching phrase was found, the phrase was selected and the simulation ended. If no matching phrase was found, however, we tested three interaction strategies. The first strategy (Strategy 1) invoked the KeywordAE (Fig. 2) iteratively by spelling more of the words out, until the matching phrase was found. The second strategy (Strategy 2) was identical to Strategy 1, except FillMask (Fig. 3) was used in lieu of KeywordAE whenever there remained only a single incorrect word in the best-matching phrase candidate. The flowcharts for Strategies 1 and 2 are shown in Fig. 4A, B, respectively. The third strategy, referred to as Strategy 2A, was a variant of Strategy 2. It utilised FillMask more aggressively, i.e. as soon as two or fewer incorrect words remained in the best option. In all three strategies, KeywordAE was invoked incrementally by spelling out more words in the best-matching candidate. This incremental spelling was implemented differently for the two versions of KeywordAE, due to differences in what abbreviations were supported. For KeywordAE v1, which supported only fully-spelled keywords, the simulation spelled out the first incorrect word in the best option; for v2, in which keywords could be partly spelled, the simulation added one additional letter at a time in the first incorrect word. To leverage the contextual understanding of the AE and FillMask LLMs, all the previous turns of a dialogue from the TDC corpus were used for expanding abbreviations and finding alternative words, unless otherwise stated.

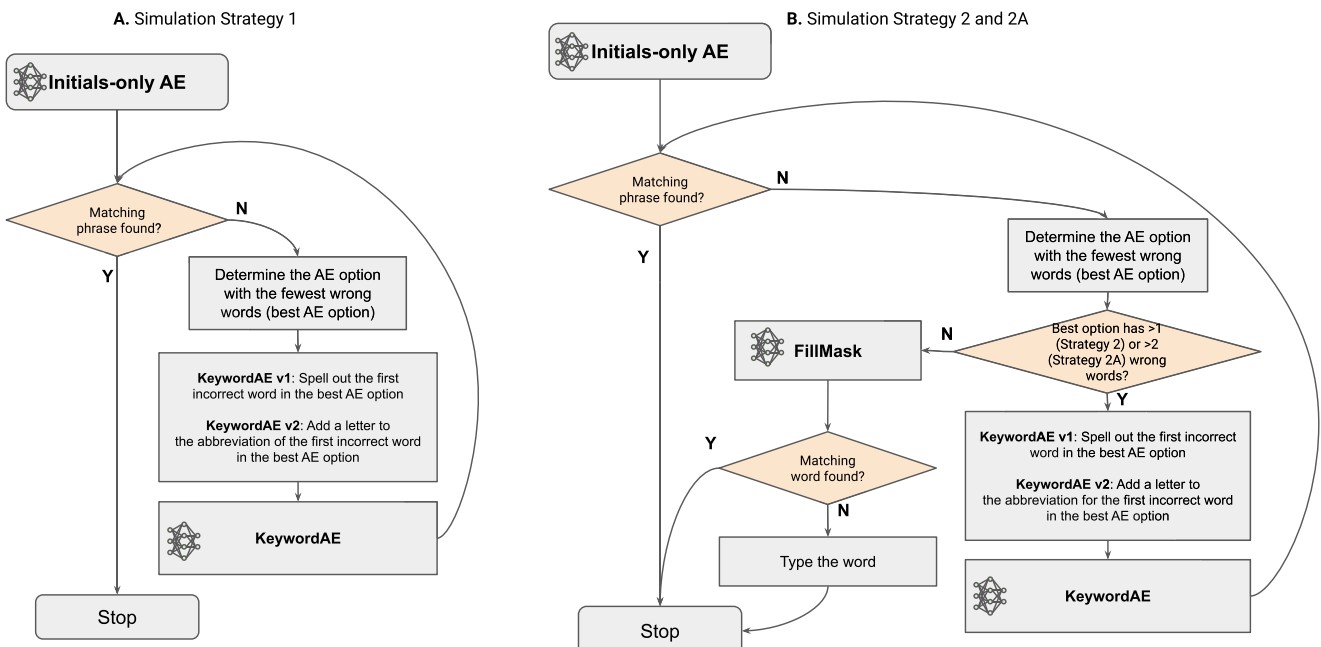

**Fig. 4 | Simulation strategies of phrase entry assisted by the Keyword Abbreviation Expansion (KeywordAE) and FillMask LLMs. A** Simulation Strategy 1: AE with only initials is followed by KeywordAE if the initials-only AE doesn't return the desired phrase. Keyword AE in v1 iteratively spells out the first (leftmost) words in the best-matching phrase option until the desired phrase is found. KeywordAE in v2 iteratively appends letters to the first (leftmost) incorrect word in the best-matching phrase option. **B** Simulation Strategy 2: same as Strategy 1, except that FillMask is used whenever only one incorrect word remains in the best-matching phrase.

As a baseline for comparison and to emulate the traditional n-gram-based text prediction paradigm, we also ran simulations on the same corpus with an n-gram LM (Gboard's finite state transducer[35] trained on 164,000 unigrams and 1.3 million n-grams in US English) that supported word completion and prediction, modelling ideal user behaviour that selects the intended word as soon as it became available among the top-n word completions or prediction options.

To quantify the number of motor actions, we broadened the definition of keystrokes to include not only keypresses on the keyboard but also UI actions required to use the phrase- and word-prediction features in SpeakFaster, including entering the KeywordAE mode (a click to the 'Spell' button in Fig. 2), specifying a word to spell for KeywordAE, entering the FillMask mode, selecting a word to replace through FillMask, and selecting phrase and word options returned by the LLMs. Similarly, the number of keystrokes in the Gboard simulation included selecting options from word completion and next-word prediction.

The result of the SpeakFaster simulations indicated a significant saving of motor actions compared to the baseline from Gboard's forward predictions (Fig. 5A). This held true for both KeywordAE v1 and v2. Under KeywordAE v2, given that SpeakFaster utilised all previous dialogue turns as the context and provided five options at each step (orange bars in Fig. 5A), Strategy 1 and Strategy 2 led to the keystroke-saving rate (KSR) values 0.640 and 0.657, respectively, significantly exceeding the Gboard KSR (0.482). These KSRs from the KeywordAE v2 model also beat the best KSR from KeywordAE v1 (0.647) by a small but noticeable margin, reflecting a benefit of allowing keywords to be partially spelled. The superior KSR of Strategy 2 relative to Strategy 1 indicates a benefit of augmenting KeywordAE with FillMask, which surfaced the correct word options with fewer motor actions required. However, the comparison between Strategy 2 and Strategy 2A shows that FillMask negatively impacts motor saving rate if used too aggressively. Specifically, premature uses of FillMask, i.e. whenever two incorrect words remained (instead of one incorrect word as in

Strategy 2), reduced the KSR from 0.657 to 0.647 (Fig. 5A). In Fig. 5A, the grey bars show that SpeakFaster outperformed the Gboard baseline in KSR even without utilising the previous dialogue turns as context, although the KSR gains were significantly lower compared to if the context was utilised.

The results in Fig. 5A are all based on providing 5-best phrase options from the KeywordAE and FillMask LLMs. To illustrate the effect of varying the number of LLM options, Fig. 5B plots the KSRs against the number of LLM options. Similar to the trend from Gboard, KSRs in SpeakFaster increased monotonically with the number of options, but started to level off at approximately five, which forms the basis for our UI design decision of including 5-best options (Fig. 1). Fig. 5C shows that when conversational context was made available to the KeywordAE LLM (either v1 or v2), approximately two-thirds of the dialogue turns in the test corpus could be found with only the initials-only AE call (i.e. a single LLM call). This fraction became approximately halved when the conversational context was unavailable, which again highlights the importance of conversational context to the predictive power of the LLMs.

The simulation results above show that the theoretical motor-action saving afforded by context-aware AE and FillMask surpassed that of the traditional forward prediction by 30–40% (relative). This result builds on the previous AE LLM in ref. 33 and goes a step further by supporting abbreviations that include spelled words (KeywordAE) and suggesting alternative words (FillMask), which removes 'dead ends', thus allowing any arbitrary phrase to be entered. However, as shown by prior studies[15,24,32,36], the predictive power of motor action-saving features in text-entry UIs is often offset by the added visual and cognitive burden involved in using these features, besides human errors such as misclicks and misspellings. Our system additionally involved network latencies due to calls to LLMs running in the cloud (see Supplementary Section 1.3). Therefore, the practical performance of the LLM-powered AE-based text-entry paradigm in SpeakFaster must be tested with empirical user studies. To this end we conducted a

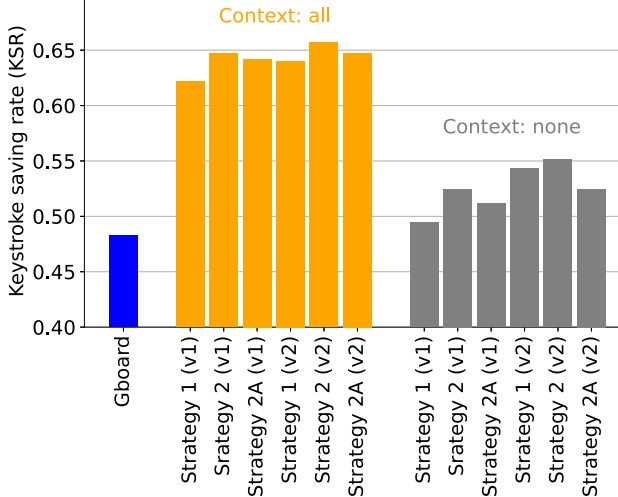

A. Keystroke saving rate (KSR) under different LLMs and simulation strategies

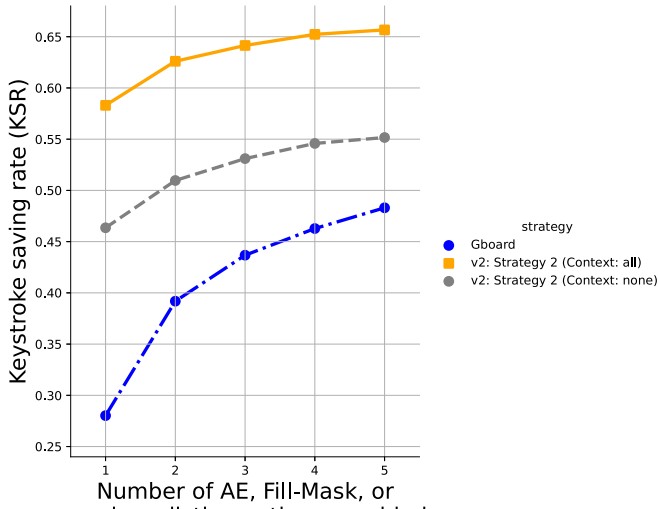

B. Keystroke saving rate (KSR) vs. number of options

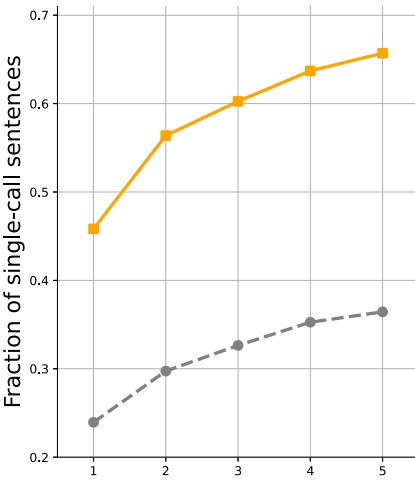

C. Fraction of turns requiring only a single LLM call

**Fig. 5 | Simulation results show significant motor savings in the SpeakFaster UI.** These results are based on the simulation strategies presented in Fig. 4 and utilising full or no conversation context as shown. **A** Keystroke saving rates (KSRs), compared with a forward-prediction baseline from Gboard (blue bar). The orange bars show the KSRs when conversational context is utilised, while the grey bars show the KSRs without utilising conversational context. All data in this plot are based on 5-best options from the KeywordAE and FillMask LLMs. **B** The KSRs from Strategies 2 as a function of the number of LLM options, in comparison with Gboard forward prediction. **C** Fraction of dialogue turns successfully entered with initials-only AE call, as a function of the number of options provided and the availability of conversational context. Each data point in this figure is the average from a single simulation run on the dialogue turns for which the sentence length was 10 or shorter (counting words and mid-sentence punctuation) from the 280 dialogues in the test split of the Turk Dialogues (Corrected) dataset[33,34].

controlled lab study on a group of users typing manually on a non-AAC mobile device as a pilot study of the novel text entry paradigm, followed by lab and field studies on two eye-gaze typing users with ALS.

**User study overview**

We tested the SpeakFaster text-entry UI with two groups of users. First, a group of non-AAC touch-typing users typed with their hands on a mobile touch-screen device running SpeakFaster powered by the KeywordAE v1 and FillMask LLMs. In a separate AAC eye-gaze user study, users with ALS who were experienced eye-gaze typers entered text by using eye trackers integrated with SpeakFaster. In all our studies, participants only viewed the partner conversations and typed their conversation turns. Except in the AAC eye-gaze field user study, where users participated in conversations in their natural environment, all partner conversations for the other AAC and non-AAC user studies were presented as text to the user. The non-AAC user study

served as a pilot for the AAC eye-gaze user study by proving the learnability and practicality of the LLM-based text-entry UI. A common goal of the two studies was to understand the cognitive and temporal cost introduced by the SpeakFaster UI and how that affects the overall text-entry rate compared to a conventional baseline. To study this under different levels of spontaneity and authoring task load, our study protocol consisted of both a scripted phase and an unscripted phase.

The scripted phase consisted of 10 dialogues from the test split of the TDC corpus[33,34]. Each dialogue is a two-person conversation with a total of six turns, three uttered by each person. Our user study participant played the role of one of the persons in the conversation, and the to-be-entered text was displayed to them close to their typing area. In the unscripted phase, the user engaged in a set of five six-turn, text-based dialogues with the experimenter where only the starting question was predetermined, and the rest was spontaneous but required

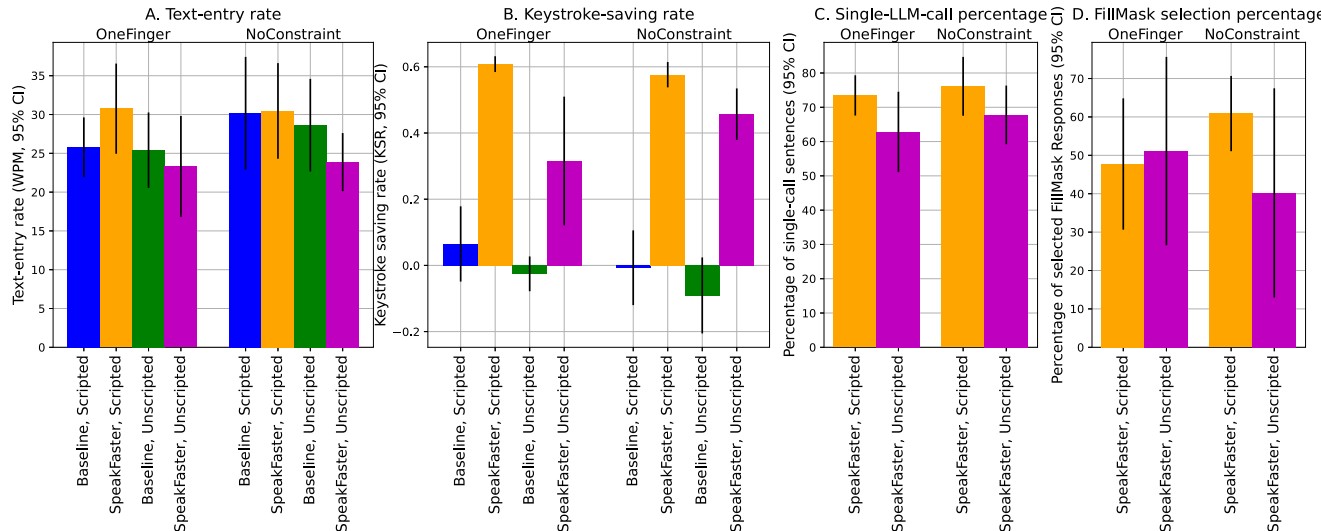

**Fig. 6 | Non-AAC user study results show improved keystroke-saving rate (KSR) with mixed changes in text-entry rate.** The results shown are text-entry rate (**A**), KSR (**B**), the percentage of sentences that involved only a single AE server call (**C**) and the percentage of FillMask responses that contain options selected by users (**D**) from the non-AAC user study. In each plot, the two groups of bars correspond to the two posture groups (OneFinger and NoConstraint), as labelled at the top. The error bars in these plots show 95% confidence intervals (CIs) of the mean. The numbers of subjects are 9 and 10 for the one-finger and no-constraint groups, respectively. This user study was conducted once. Overall, the text entry speed (WPM) using SpeakFaster was not significantly different from the baseline despite introducing significant keystroke savings.

the user to keep each conversation turn under ten words and not include any personal information. The experimenter, who entered text on a custom mobile app connected with the study participant's device, started with an open-ended question such as 'What kind of music do you listen to?' (see Supplementary Section 1.4 for the full list) and then the user would reply. The experimenter and user then followed up in alternating turns, until a total of six turns was reached for each dialogue. In both the scripted and unscripted phases, for each block of five dialogues, the first and last dialogues formed the baseline, in which the user typed with the regular keyboard i.e. either Gboard in the non-AAC user study or the Tobii eye-gaze keyboard in the AAC eye-gaze users study, utilising word suggestions provided by the default keyboard at will. In the other three dialogues, the user entered text by using the SpeakFaster UI, starting out with the initials-only abbreviation scheme and using any of the pathways to either spell out words (KeywordAE) or select replacements (FillMask) as per their preference until they were satisfied.

Prior to the data collection portion of the lab study, each user watched a video demonstration of the SpeakFaster system and took a short practice run to familiarise themselves with the SpeakFaster UI and the AE text entry paradigm. Each non-AAC user study participant practiced a minimum of five practice dialogues. The content of these practice dialogues was different from the ones used in the subsequent study blocks. Prior to the unscripted phase, the users were also familiarised with the unscripted test condition through practicing a single dialogue. The eye-gaze user practiced for 4 h over 2 days and this is described more in the Supplementary Section 1.4.

### Non-AAC users' SpeakFaster text-entry rates are similar to baseline

In order to study abbreviated text entry under different degrees of motor cost, the 19 participants (10 female, 9 male, all adults) who provided informed consent were randomly assigned to two typing-posture groups in the non-AAC user study. Nine users were assigned to the one-finger group and instructed to type with only the index of their dominant hand (right hand for all these users). The remaining ten users were assigned to the no-constraint group and were given no limitation

related to typing posture. They all operated with both hands during the experiment, with varied posture details in which fingers or thumbs were used.

In the scripted portion of the user study, no significant difference was observed in the accuracy of text entry between SpeakFaster and the Gboard baseline. The average word error rates (WERs) of the OneFinger group were 1.55% and 2.53% under the baseline and SpeakFaster conditions, respectively. For the NoConstraint group, the respective average WER were 3.96% and 2.89%. A two-way linear mixed model (Posture × UI) on the WERs showed no significant main effect by Posture ($z = -1.758$, $p = 0.079$) or UI ($z = 0.079$, $p = 0.250$). Nor was there a significant interaction in WER between Posture and UI ($z = 1.516$, $p = 0.129$).

The effect on text-entry rate by the LLM-powered SpeakFaster UI showed an intricate mixed pattern. While the text-entry rate saw increases on average under the SpeakFaster UI relative to the baseline (Gboard) UI during the scripted dialogues, the average rate showed a decrease when the users engaged in unscripted ones. Analysis by a linear mixed model did not reveal a significant main effect by UI ($z = 0.141$, $p = 0.888$). However, a significant two-way interaction was found between UI and DialogType ($z = -2.933$, $p = 0.003$). Post hoc paired $t$-test (two-tailed, same below) confirmed a significant difference in the SpeakFaster-induced changes in the text-entry rate (relative to baseline) between the scripted and unscripted dialogues ($t_{18} = -4.85$, $p < 0.001$, Cohen's $d = 1.11$, 95% CI of difference, same below: $-6.00 \pm 2.53$ WPM). Specifically, while SpeakFaster increased the average rate by $2.510 \pm 3.024$ WPM (95% CI of mean, relative: $13.0\% \pm 24.5\%$ WPM) under the scripted dialogues, it decreased the average rate by $3.494 \pm 3.294$ (relative: $10.2\% \pm 25.0\%$) under the unscripted ones. The three-way linear mixed model did not reveal any other significant main effects or interactions.

### Significant motor savings under SpeakFaster

While the effect of SpeakFaster on the text-entry rate exhibited a complex pattern of interactions, showing an absence of overall significant change from the baseline, the KSR was affected in a clear-cut and pronounced manner by SpeakFaster (Fig. 6B). The same three-way linear mixed model, when applied on the KSR as the dependent

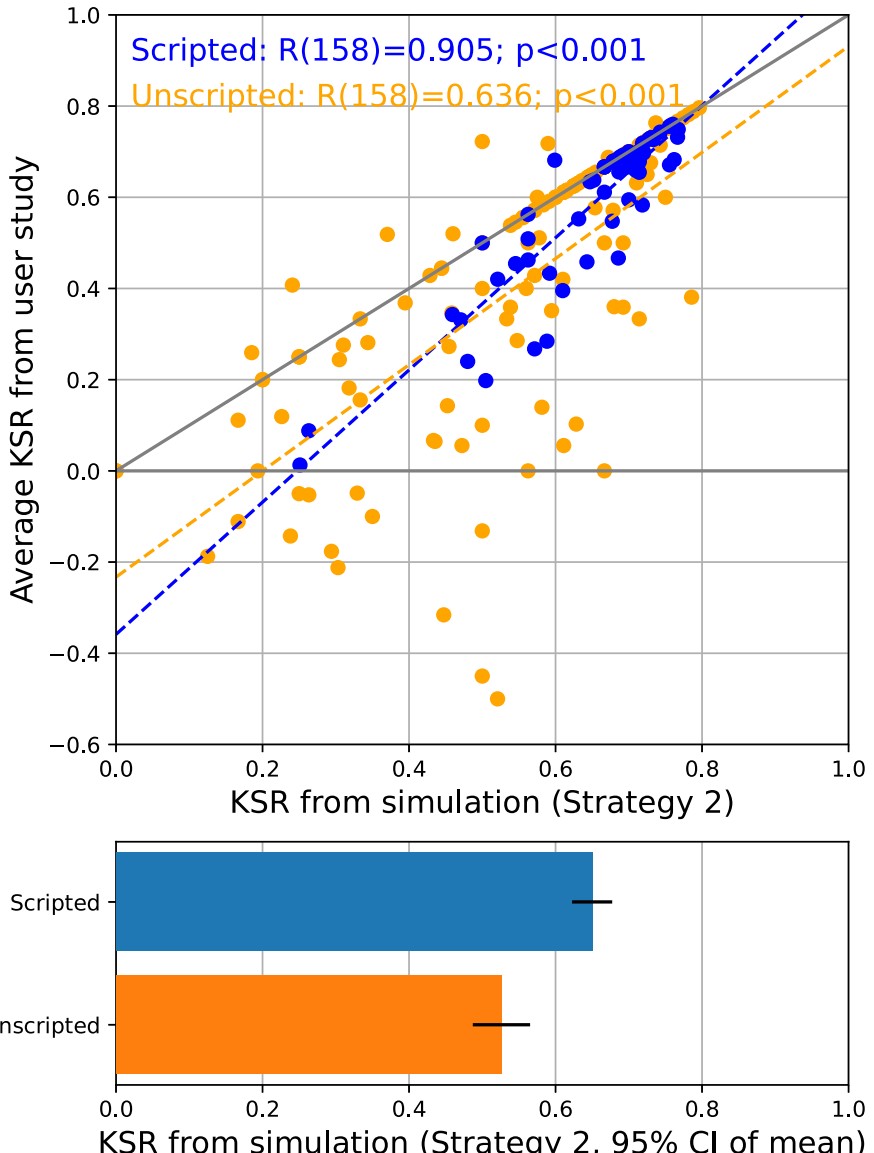

**Fig. 7 | User behaviour in SpeakFaster is well-predicted by simulation results.** The top panel shows correlations between the average KSRs observed in the users (y-axis) and those from simulation (x-axis), on a turn-by-turn basis. The blue and orange dots correspond to the scripted and unscripted dialogue turns, respectively. The simulation results are based on KeywordAE model v1, Strategy 2, five predictions and using all previous turns of the ongoing dialogue as the context.

Each dot corresponds to a unique turn from one of the dialogues used as the text materials in the lab study. The numbers of unique scripted and unscripted dialogue turns are 60 and 160, respectively. The bottom panel shows a comparison of the simulated KSR values from the scripted (blue) and unscripted (orange) dialogue turns. The KSRs from the scripted turns were significantly higher than those from the unscripted ones (unpaired $t$-test: $t_{218} = -3.656$, $p < 0.001$).

variable, revealed a significant main effect by UI ($z = 10.317$, $p < 0.001$). The linear mixed model revealed no other significant main effects or interactions. Relative to the Gboard baseline, the SpeakFaster UI paradigm led to a large and significant increase in KSR for both the scripted dialogues ($+0.564 \pm 0.080$ abs., $t_{18} = 13.44$, $p < 0.001$) and the unscripted ones ($+0.450 \pm 0.114$ abs., $t_{18} = 7.556$, $p < 0.001$).

Panel C of Fig. 6 shows the percentage of dialogue turns in which the user successfully entered the sentence by using only the initials-only AE call, i.e. without spelling out words in the abbreviation or using FillMask. As the orange bars show, the percentages were on par with the results from the simulation in the scripted dialogues (c.f. Fig. 5C). The percentages of sentences that succeeded with a single AE call were lower for unscripted dialogues (magenta bars, 65% on average), reflecting the slight domain mismatch in the unscripted text content from the scripted ones that the AE and FillMask models were trained on.

### Simulation accurately predicts users' keystroke savings
The KSR values observed from the users in the lab study could be predicted with high accuracy by the simulation results. The blue dots in the top panel of Fig. 7 show a significant positive correlation between the average KSR values from all users and the simulated ones on a turn-by-turn basis among the scripted dialogues (Pearson's correlation: $R_{158} = 0.905$, $p < 0.001$). The unscripted dialogues (orange dots) also exhibited a significant correlation between the simulated and observed KSRs ($R_{158} = 0.636$, $p < 0.001$). However, it can be seen that the users' average performance did not fully realise the motor-saving potentials predicted by the offline simulations, as most data points in Fig. 7 fall below the line of equality (the solid diagonal line), potentially reflecting human errors such as typos and mis-operations, as well as the actions needed to recover from them. The degree to which the user behaviour underperformed the simulation results was greater during the unscripted dialogues than the scripted ones. For

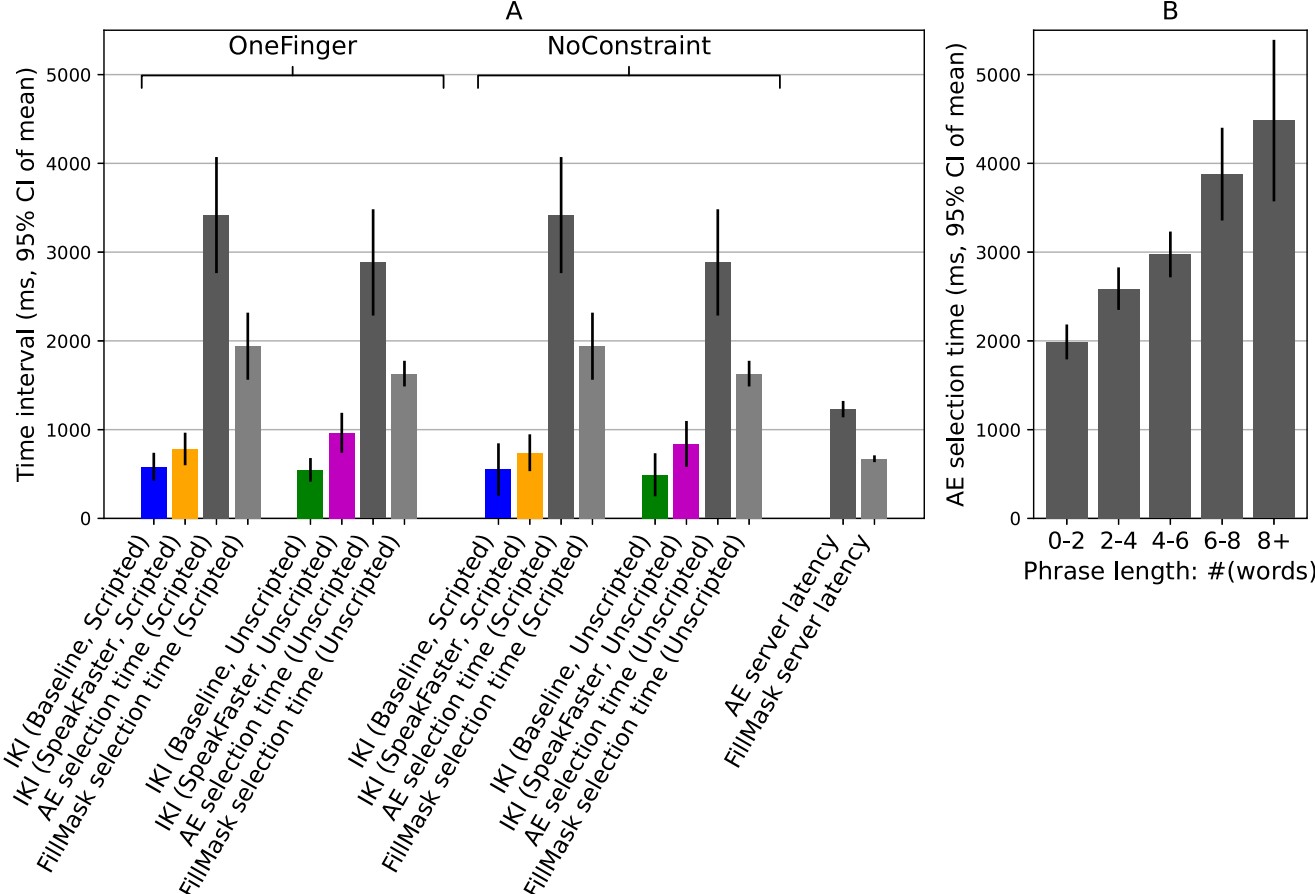

**Fig. 8 | Evaluating and selecting LLM-provided options consumes a large fraction of the time during SpeakFaster UI usage.** Temporal analyses of text entry in the SpeakFaster UI and the Gboard baseline in the non-AAC user study. **A** The inter-keystroke intervals (IKIs), response times for selecting AE and FillMask (FM) options returned from the LLMs, and the latencies of calls to cloud-based LLMs. The bars are organised into five groups. The first two groups are from the OneFinger posture group, showing scripted and unscripted dialogues, respectively. Likewise, the third and fourth groups show the results from the NoConstraint posture group. The fifth group shows LLM latencies. **B** the response time for AE option selection showed a clear increasing relation with the length of the underlying phrase, measured as a number of words defined as the character length of the phrase divided by 5. The numbers of subjects in the one-finger and no-constraint groups are 9 and 10, respectively. The rightmost two bars in (**A**) and the bars in (**B**) are computed on the pooled group of 19 participants.

example, several unscripted dialogue turns showed negative KSRs, despite the fact that offline simulation predicted positive KSRs based on the final committed sentences. This likely reflects greater revisions due to human errors and change of mind when the users operate under the dual cognitive load of formulating a dialogue response and operating the SpeakFaster UI in order to enter the response.

The bottom panel of Fig. 7 shows that the simulated KSR values were significantly lower for the unscripted dialogues than the scripted dialogues (mean: 0.527 vs. 0.650, −18.99% relative, unpaired $t$-test: $t_{218} = -3.656$, $p < 0.001$, Cohen's $d = 0.553$, 95% CI of difference: −0.124 ± 0.067). This was likely due to a domain mismatch between the TDC dataset and the unscripted content composed by the users during the lab study. However, the fact that SpeakFaster significantly boosted KSR even for the unscripted dialogues (Fig. 6) underlines the robustness of the motor saving against domain shifts.

**Temporal aspects of user interactions in SpeakFaster**
Figure 8 shows the temporal aspects of how the non-AAC users interacted with the SpeakFaster UI. An inter-keystroke interval (IKI) is defined as the time interval between two consecutive keystrokes issued directly by the user using the soft keyboard (Gboard). IKIs exclude non-keystroke actions and the keystrokes are automatically applied due to the user selecting options for word completion and predictions. The IKIs were significantly longer under the SpeakFaster

UI than the baseline UI, showing that it was slower on average for the users to plan and perform the keystrokes when typing the characters of the abbreviations and performing the keystrokes required for subsequent spelling of words than typing words in the familiar, sequential fashion (main effect by UI: $z = 3.671$, $p < 0.001$). The linear mixed model also identified a significant UI × DialogType interaction ($z = 2.303$, $p = 0.021$). A post hoc $t$-test confirmed that the increase in the IKI was greater under the unscripted dialogues (+381.1 ms abs., 87.1% rel.) than under the scripted ones (+194.8 ms abs. 51.7% rel., $t_{18} = 3.066$, $p = 0.0067$, Cohen's $d = 0.703$, 95% CI of difference: 186.366 ± 124.2 ms). Similar to the observations related to KSRs above, this differential effect of SpeakFaster on IKI increase may be interpreted as the greater cognitive load under the dual task of composing a free-form dialogue response and abbreviating the response in the SpeakFaster's abbreviation regime.

The dark and light grey bars in Panel A of Fig. 8 show the temporal metrics related to using the LLMs for the AE (including KeywordAE) and FillMask workflows, respectively. Compared with the latencies of the calls to the cloud-based LLMs (two rightmost bars in Panel A), the users' average response times in selecting the AE-suggested phrases and FillMask-suggested words were 2–3 times longer. These response times not only exceeded the LLM latencies, but were also longer than the average IKIs by 3–6 times, indicating that they were a significant component of the total time it took to use the SpeakFaster UI.

The average response times for AE were approximately twice as long as those for FillMask. This is attributable to the fact that the AE options are multi-word phrases while the FillMask options are single words, which highlights an additional benefit of the word-replacement interaction in FillMask. As Fig. 8B shows, the AE response time showed a strong positive correlation with the length of the phrase (Spearman's $\rho_{423} = 0.428$, $p < 0.001$). While selecting phrase options of length two words or shorter took only ~2000 ms on average, selecting phrases eight words or longer took more than twice as long.

The results of the non-AAC user study showed that the LLM-based SpeakFaster text entry UI led users to achieve savings in motor actions including keystrokes up to 50 percentage points (absolute) higher than the conventional way of mobile text entry. In terms of speed, the results were mixed. While in the scripted dialogue condition the users achieved an average of 13% speedup, they showed a 10% slowdown under the unscripted condition, reflecting an interplay between the added cognitive load of using the UI and that required to compose a spontaneous text message. Timing analysis revealed that reviewing the phrase options from the KeywordAE LLM took about 3–6 times the average IKI; it took relatively less time to review the word options in FillMask, but the review cost was still significant (2–3 times as long as the average IKI). These timing findings highlight an important trade-off between the cost of reviewing LLM outputs and the savings in motor actions. Mobile touch-typing IKIs were relatively short (≈500 ms, Fig. 8A), which may mask the benefit from the predictive power of the LLMs. However, in eye-gaze typing, the IKIs can be significantly longer, owing to the dwell time and the gaze travel time between keys. These considerations indicate stronger potentials for acceleration in eye-gaze text entry than mobile text entry. With this hypothesis, we proceeded to study AAC eye-gaze users' interaction with SpeakFaster.

### SpeakFaster enables significant KSR and WPM speed-up for eye-gaze users

The two study participants were both adult males diagnosed with ALS (17 and 11 years prior to the study, respectively) and provided informed consent before participating in this study. Both users were native speakers of American English and experienced eye-gaze typists who communicate daily through eye trackers and associated keyboards. At the time of the study, both participants were quadriplegic, unable to speak, however, their eye movements remained functional and their cognitive abilities were reported within normal limits. They were experienced with the Tobii ® eye-gaze on-screen keyboard and its n-gram word competition and prediction features similar to Gboard. The first participant engaged in a controlled lab study (LP1) and the second in a field deployment to gather more naturalistic data (FP1).

In the controlled lab study, our participant (LP1) followed a structured study protocol consisting of a scripted part followed by an unscripted part identical to that of the mobile user study described above. However, LP1 used the KeywordAE v2 model, which supported partially-spelled keywords and triggered LLM calls for initials-only AE and KeywordAE after every eye-gaze keystroke, eliminating the need for gaze trigger of AE LLM calls as in the non-AAC user study. Prior to the data collections, the user practiced the SpeakFaster text entry paradigm under the direction of the experimenter for a total of 4.1 h over two separate days before the lab study, for which the learning curve can be found in Supplementary Section 1.4.

Figure 9 A compares the mean text-entry rates in WPM between the SpeakFaster paradigm with the Tobii keyboard baseline. Averaged over the scripted dialogues, the user achieved an average text-entry speed of 6.54 WPM while using SpeakFaster, which exceeded the baseline typing speed (4.05 WPM) by 61.3% (two-sample $t$-test: $t_{28} = 2.76$, $p = 0.010$, Cohen's $d = 1.03$, 95% CI of difference: 2.484 ± 1.841 WPM). A similar rate enhancement occurred for the unscripted dialogues (SpeakFaster: 6.38 WPM, baseline: 4.37 WPM, 46.4% increase), although the difference did not reach significance

($t$-test: $t_{13} = 0.818$, $p = 0.43$, Cohen's $d = 0.575$). In addition to increased speed, a significant increase in the KSR was also observed for the user with SpeakFaster for both the scripted and unscripted dialogues. Again, statistical significance was reached only for the scripted dialogues (0.360 vs. the baseline of 0.227, rank-sum test: $\rho = 1.97$, $p = 0.049$, Cohen's $d = 0.575$, 95% CI of difference: 0.133 ± 0.177, Fig. 6B).

In the lab study, 77.8% of the scripted dialogues and 66.7% of the unscripted ones required only a single initials-only AE LLM call. For the trials in which LP1 used FillMask, the LLM predicted the correct words (as determined by user selection) 58.3% of the time for the scripted dialogues and 21.4% of the time for the unscripted ones. Despite achieving success a majority of the time (58.3%), the FillMask LLM's success rate observed on LP1 on the scripted dialogues is lower than the success rate predicted from offline simulation (70.7%), indicating that the user occasionally failed to choose the correct words when they appeared. The fact that the success rate of the initial AE and FillMask was lower for unscripted dialogues reflects the domain mismatch between the user's personal vocabulary and the model's training corpus, highlighting personalisation of the model as a useful future direction.

To study the efficacy of the SpeakFaster paradigm under more natural usage, we conducted a field study with another eye-gaze user (FP1). As we reported previously[37], FP1 showed an average eye-gaze text-entry speed of 8.1 ± 0.26 WPM (95% CI) over a period of 6 months of measurement in his real-life communication with close relatives and caregivers. This baseline speed of gaze typing is based on 856 utterances typed with the Tobii ® Windows Control eye-gaze keyboard with a PCEye Mini IS4 eye tracker. FP1 also engaged in test dialogues with an experimenter by using the SpeakFaster UI based on KeywordAE v1 and manual triggering of LLM calls for AE. Over the 27 unscripted phrases entered with SpeakFaster on six different days, the user achieved an average speed of 10.4 ± 2.6 WPM, which is 28.8% faster than the daily baseline (two-sample $t$-test: $t_{881} = 2.97$, $p = 0.0031$, Cohen's $d = 0.580$, 95% CI of difference: 2.335 ± 1.544 WPM, Fig. 9C). Accompanying this increase in the average speed of text entry was an increase of KSR from −0.14 to 0.32 (rank-sum test: $\rho = 4.37$, $p < 0.001$, Cohen's $d = 0.566$, 95% CI of difference: 0.463 ± 0.314, Fig. 9D).

### Motor savings outweigh cognitive overhead for eye-gaze typing

The latencies for the AE calls were 843.0 ± 55.4 ms and 832.7 ± 120.4 ms (95% CI of mean) for the scripted and unscripted dialogues, respectively (Fig. 10A). The latencies of the FillMask calls were shorter (scripted: 617.9 ± 41.8 ms; unscripted: 745.2 ± 67.1 ms) than AE, due to its serving configuration that took advantage of the shorter output lengths (Supplementary Section 1.3). These LLM-serving latencies were approximately four times shorter than the average eye-gaze IKIs measured on user LP1 (Fig. 10A, blue and orange bars, 3511–4952 ms) and therefore had only a minor slowing effect on the text-entry rate of AAC eye-gaze typers. In comparison, the time it took the user to select the correct AE responses was significantly longer (Fig. 7B: scripted: 12,732 ± 5207 ms; unscripted: 21,225 ± 19,807 ms), which was 3–6 times the average duration of a keypress, reflecting the significant cost for scanning the phrase options from the AE calls. By contrast, FillMask involved a much shorter (2–3×) candidate selection time than AE (Fig. 7B: scripted: 7032 ± 4584 ms; unscripted: 4745 ± 2023 ms), reflecting the benefit of the FillMask interaction in providing shorter, single-word candidates, which reduced the scanning cost.

Compared with the average IKI in the non-AAC users (Fig. 8), these IKIs from the eye-gaze typist shown in Fig. 10A were 3–6 times longer. This provides insight as to why SpeakFaster leads to a significant speed-up for eye-gaze typists in this study while introducing minimal changes among the participants in non-AAC user study described above. Specifically, Panel B of Fig. 10 shows a breakdown of the time intervals spent on several different types of actions during the baseline typing condition and SpeakFaster typing

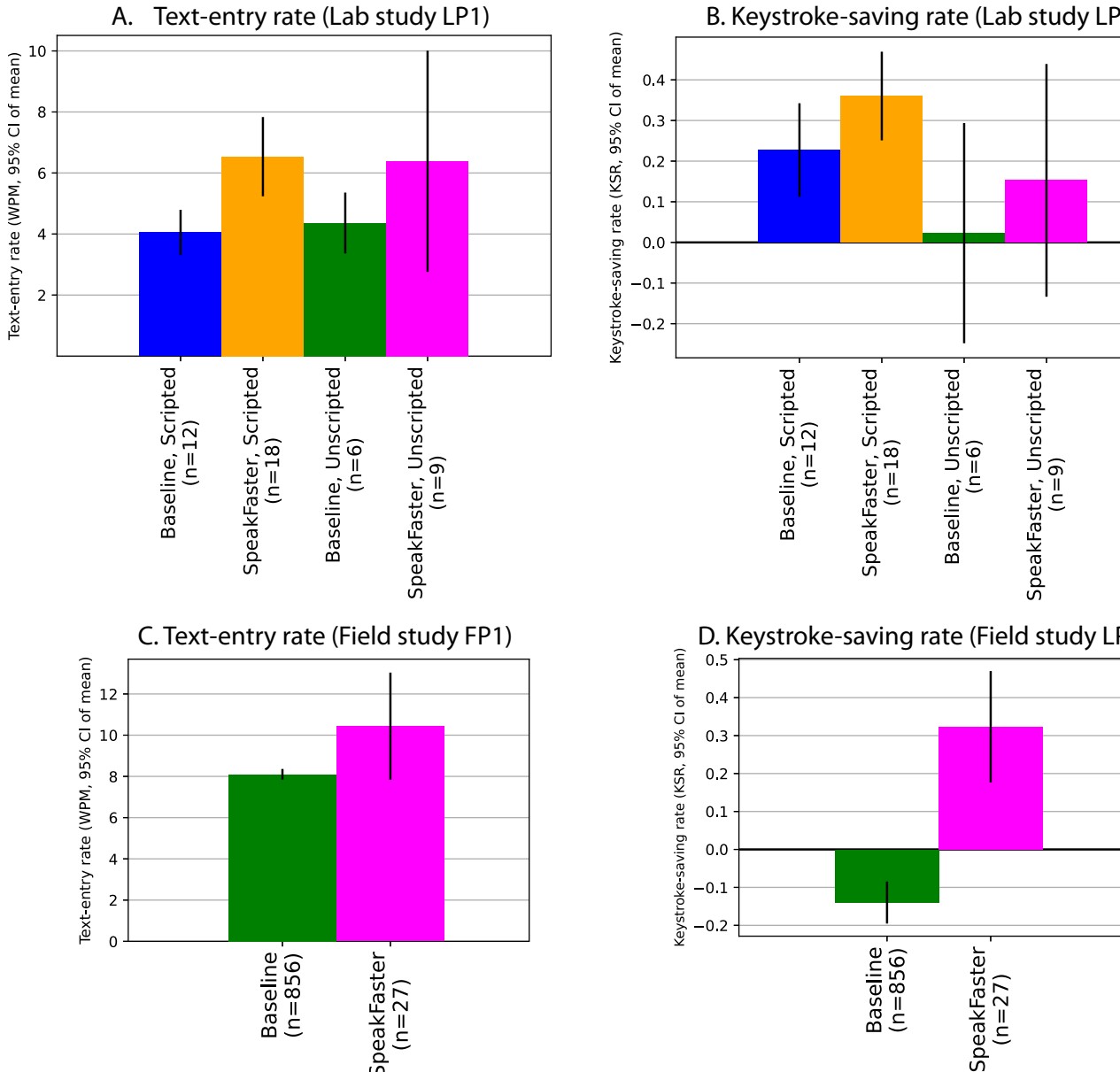

**Fig. 9 | AAC eye-gaze users' text-entry rates and KSR show gains under the SpeakFaster testing condition compared to a non-LLM-assisted baseline.** **A**, **B** Show the comparisons of the text-entry speed and keystroke saving rate (KSR) between SpeakFaster's AE UI and a forward prediction baseline based on the Tobii eye-gaze keyboard in the lab study participant LP1. **C**, **D** Compares the text-entry rate and KSR of the field study participant FP1 while using SpeakFaster versus a Tobii keyboard baseline, for both scripted dialogues and unscripted dialogues in a controlled lab study. Error bars show 95% confidence interval (CI) of the mean over the numbers of dialogue turns in the lab and field studies, each of which was conducted once respectively, as indicated in the x-axis labels. Note the different y scales between the panels.

condition, as shown by different colours. The overhead of using SpeakFaster is broken down into the LLM latencies, the actions involved in using KeywordAE (entering the spelling mode, selecting a word to spell, as well as reviewing and selecting phrase options), and those involved in using FillMask (entering the FillMask mode, selecting the target word, as well as review and selecting word options). Among these subtypes of overhead, the LLM latencies were a relatively minor factor. The total overhead of using the SpeakFaster system is 5215 ms/word for the AAC users and 1735 ms/word for the non-AAC users. While this timing overhead is close to the time spent on keystrokes for the non-AAC users (2252 ms/word); it was much shorter than the time spent on keystrokes for the AAC eye-gaze user

(who is at 14,774 ms/word) leading to a reduction in the overall time on a per word basis.

By contrast, the non-AAC users only showed a more modest average reduction of time spent on keystrokes (1416 ms/word) when using SpeakFaster, which was insufficient to fully offset the increased overhead. This was largely due to the fact that the average IKIs were already much shorter during mobile typing than in eye-gaze typing.

In summary, the quantitative measurements of eye-gaze typing in both users found a considerable advantage of the SpeakFaster UI of text input compared to the baseline of conventional eye-typing systems with forward prediction. The evidence is seen both in a controlled lab setting and a field study in comparison with a real-life, long-term baseline.

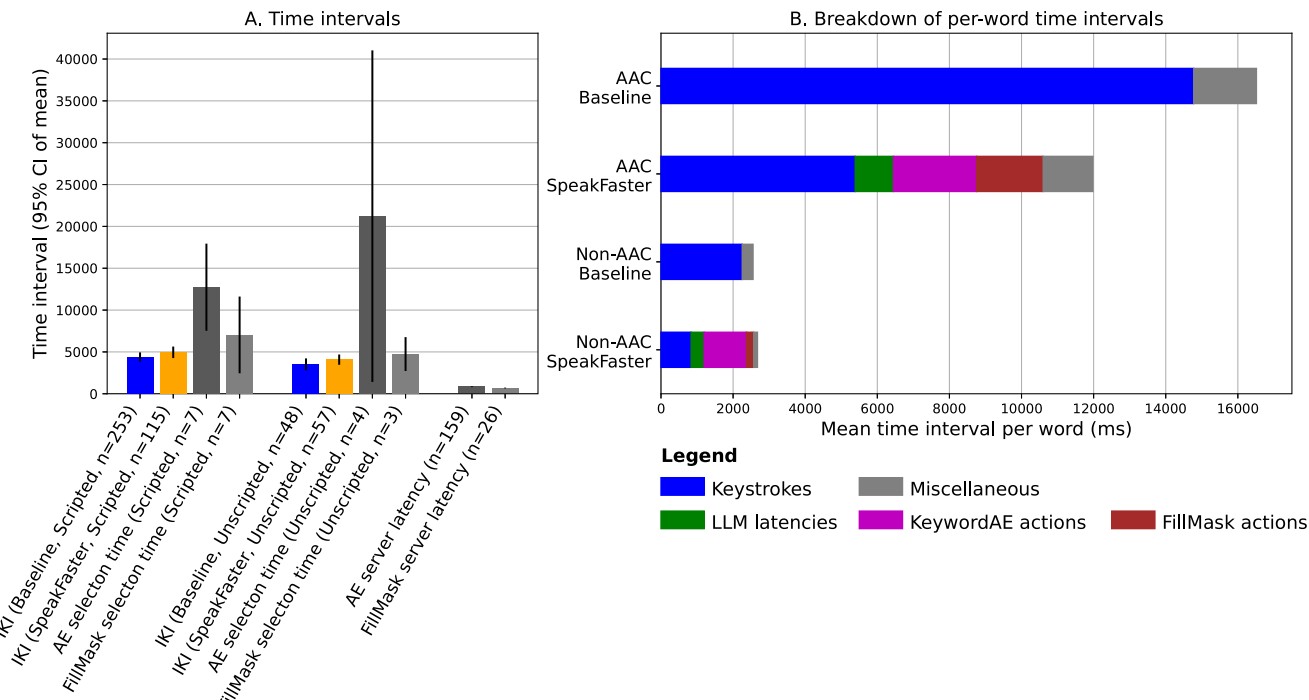

**Fig. 10 | Savings in motor actions outweigh the cost of LLM option evaluation in eye-gaze AAC users.** Comparison of inter-keypress intervals, server latencies, and user-selection reaction times observed in the lab study on user LP1.
**A** Compares the inter-keystroke intervals (IKIs) of LP1's eye-gaze keypresses, the time it took the user to select LLM-provided options for KeywordAE and FillMask, and the server latencies (including network latencies) for the LLM calls. The error bars show 95% of the mean over the number of events as specified in the x-axis labels. **B** Shows breakdown of the amount of time spent on different tasks under the baseline and SpeakFaster typing conditions, observed in the eye-gaze user LP1 (two bars at the top) and the average over the 19 mobile-study users (two bars at the bottom). Only the data from the scripted typing condition are shown for clarity. The colour code of the bar segments is omitted in the bottom two bars due to space limit, but is identical as that of the top two bars.

## Discussion

AE of phrases in user-input history has long existed as a feature of eye-gaze AAC software, in addition to being seen in non-AAC applications such as text messaging, and hence is familiar to users of such systems. SpeakFaster builds on this user knowledge, while drastically extending its breadth to open-ended phrases including previously untyped phrases by using contextualised predictions from fine-tuned LLMs. Our demonstration of accelerated text entry focused on eye-gaze typing for users living with ALS. However, the same LLM-based paradigm can be extended to other modalities of text entry in the AAC realm. For instance, SpeakFaster's AE and FillMask design should serve to reduce the number of scans and selections required for text entry based on switch scanning, a type of UI suitable for users who are unable to use eye tracking devices due to more pronounced motor impairments[38]. Similarly, BCI-based text entry, including techniques based on motor imagery from surgical implants[11,39] and the non-invasive techniques based on visual evoked potentials[14] could benefit from the same greater predictive power of LLMs. Furthermore, for users without disabilities, constraints such as small screens and situational impairments[40,41] may also render the temporal and motor cost of text entry high, in which case it will be interesting to explore the LLM-based techniques such as shown in the current study for motor savings and temporal acceleration. Such envisioned applications of the SpeakFaster paradigm remain to be investigated and confirmed in future studies.

### Relation to C-PAK (Correcting and completing variable-length Prefix-based Abbreviated Keys) paradigm
Our abbreviation-based writing scheme can be viewed as an implementation of the C-PAK (Correcting and completing variable-length Prefix-based Abbreviated Keys) text-input paradigm[36]. Compared to traditional forward predictions, C-PAK further exploits the redundancy and compressibility of text through the realisation that the information that helps predict a word is present not only in its preceding words (left context), but also in the words that follow it (right context) even if these words are only partially spelled out, an insight shared with bidirectional neural LMs such as BERT[26]. PhraseWriter, the first implementation of C-PAK, used count-based n-gram LMs with narrow context windows. In a lab study, users of PhraseWriter showed lower average text-entry rate than a non-abbreviation-based Gboard baseline, which was attributable to the limited AE accuracy and therefore frequent revisions through backspace and re-typing[36].

To our knowledge, this study demonstrated acceleration of text entry in the C-PAK paradigm for the first time through the enhanced text entry rates, i.e. in the two eye-gaze users. The results in this study also push forward the Pareto frontier of the KSR-accuracy tradeoff relation demonstrated by Adhikary et al.[42], for both typing on mobile devices and with eye gaze. This acceleration was brought about by combining the greater context awareness[17,18] of the LLM-based AE and a UI design that allows initial AE failures to be repaired without the costly use of delete or backspace keys. Featuring its affordances for spelling words and finding alternative words, the SpeakFaster UI allows users to spend a reasonably low amount of editing effort when the intended phrase couldn't be found through the initial abbreviation. It should be noted that the KSR achieved by the eye-gaze typists were in the range of 0.32–0.36. Despite significantly exceeding the KSR from their baseline typing paradigm, these KSRs still showed a significant gap from the offline simulation results (which exceeded 0.6). This was due to the unavoidable user errors when operating the SpeakFaster UI, such as mistakes and typos when entering the initials-only abbreviation and spelling out keywords, as well as mistakes when selecting phrase options from KeywordsAE and FillMask. These findings show

that future work on improving the error tolerance of the LLMs is a fruitful future direction in further improving the speed of such abbreviation-based text entry UIs.

In addition to per-word abbreviations, previous studies have also explored expanding bags of keywords into full phrases using AI (e.g. KWickChat[43]) or information-retrieval techniques[18]. Compared to these earlier designs, SpeakFaster gives users finer-grained control on exact sentence structure and wording, regardless of whether the sentence being entered has been entered before. However, two aspects of the SpeakFaster UI can be extended to bags of keywords. In particular, KWickChat can allow users to provide more keywords if the initial bag of keywords fails to yield the intended phrase; word replacement similar to FillMask also may also fit the KWickChat design.

## Length of suggestions

Can AE become a part of the conventional sequential typing, so that users can use AE when so desired and revert back to typing letter-by-letter otherwise? The answer to this question hinges on whether sentence-level abbreviations consisting of word initials (e.g. 'iipitb') can be reliably distinguished from non-abbreviated words. Another important factor to be taken into consideration is the size of the screen. Small screens limit not just the number of options that can be surfaced, but also the length of the expanded text. This can introduce secondary effects wherein editing or correcting for errors in a smaller screen can be much more demanding and difficult. Thus sentence level expansion may be even less preferred.

## Tuning vs. prompting

The current SpeakFaster system relies on two fine-tuned LLMs to reliably and accurately suggest text expansions. However, with remarkable improvements the field is seeing (e.g. GPT-4) it's conceivable that such expansions could simply be enabled by a single LLM and perhaps with no necessity for fine-tuning e.g. with prompt-tuning or even just good prompt engineering. This could make it much easier to develop SpeakFaster-like text-entry applications in the future.

## Latency and internet access

Current LLMs are large in parameter count and the amount of computation required during inference, and thus can only be served on large dedicated servers. This necessitates the mobile and AAC devices to have access to the Internet. Further, with mobile text-entry applications, latency is an important consideration. As noted before, in our case latency to query the server will be a factor. With recent advances in open source LLMs (e.g. LLAMA[44]) and software for faster inference of LLM on local devices (e.g. GGML[45]), in the near future, smaller more efficient models will be able to perform similarly. Nevertheless, at the time of writing, the application will require Wi-Fi or mobile data coverage and can piggyback off the bandwidth of existing internet-based text-messaging applications.

## Cognitive overhead

As seen in Fig. 8 the current SpeakFaster UI leads to significant cognitive overhead for users, since they face burdens from multiple aspects of the task that do not exist in traditional sequential typing. This includes the mental conversion of the intended phrase into first-letter initialism, reviewing and selecting phrase and word options predicted by the LLMs, and deciding the next action (e.g. spelling vs. FillMask). While our abbreviation scheme is still simple, the presence of contracted phrases (e.g. 'isn't', 'we'll') could take some time for users to adapt, and it would also be difficult to potentially correct abbreviations if the user didn't keep track of every word in a sentence. It remains to be seen how much user can improve their skills at this using this UI with much longer-term usage. However, the cognitive overhead appears to be within manageable bounds for our intended users, which is supported by the fact that all the users in our mobile user study and

both eye-gaze typing users with ALS were able to learn to use the UI within the duration of the study sessions (see Supplementary Section 1.5 for more details).

## Limitations and future directions

Our KeywordAE LLM used in this study did not incorporate tolerance for typos in its input abbreviations[46]. Whether such tolerance benefits the usability and text-entry rate remains to be investigated in future studies. The SpeakFaster prototype reported in this study was created for English. However, the multilingual nature of today's LLMs (e.g. LaMDA[25], PaLM[47] and OPT[48]) opens the door to extending the prototype to other languages. The extension to alphabet-based languages such as most European languages should be relatively straightforward and can be based on a similar UI design. Application on non-alphabet languages such as Chinese and languages that use mixed character sets such as Japanese are more complex and require additional considerations. However, it is worth noting that the 'Jianpin' input paradigm, where the romanised form (Pinyin) of each Chinese character is abbreviated as one or more initial letters, is similar to the SpeakFaster UI design and formed an original inspiration for the more general C-PAK paradigm.

Several limitations of the current study remain to be improved upon by future studies. First, the SpeakFaster prototype we tested limited the phrases to ten words and mid-sentence punctuation or shorter. While this length limit is sufficient to capture a majority of sentences used in daily communication, the design must be extended to longer phrase length limits to support greater flexibility and expressiveness in the future. This extension requires improvements in both the underlying LLM and UI design, as multiple candidates of longer sentences cannot fit easily onto available screen area. Second, the acceleration achieved in this study was based on the assumption that the broader conversational context (words spoken by the eye-gaze user's conversation partner) is available to the LLMs. While this approach extends the recent trend of leveraging greater context awareness in AAC research[18,37,43], the practicality of this context awareness for real-life AAC communication awaits further investigation integrating technical perspective[49] and privacy considerations[5,37]. Capturing the conversation partner's speech content requires automatic speech recognition, which has seen significant improvements in accuracy in recent years[50] and has drawn increasing interest from AAC researchers[37,49]. However, even in situations where conversation partner's turns are unavailable due to privacy and technical reasons, the previous self-entered turns by the user can still form a strong contextual signal for supporting AE and FillMask. In addition, the large size of the LLMs used in our system (64B parameters) leads to a high serving cost (16 TPU chips; Appendix Section 1.3) that may become prohibitive when scaled to a large number of AAC users in the future. Significant reductions in serving cost should be achievable through techniques such as model distillation[51] and integer arithmetic[52]. We also note that the AAC eye-gaze participants recruited for this study were an opportunistic cohort. The inclusion criteria include: (1) experienced eye-gaze AAC users who use eye-tracking AAC devices to communicate in their daily lives, and (2) report having cognitive abilities within normal limits. Hence evaluating the system with a larger group of AAC eye-gaze participants is a future direction.

Typing letter-by-letter on an on-screen keyboard by eye-gaze tracking is a non-invasive means of communication used by people with severe motor impairments such as in ALS. The SpeakFaster project showed that the efficiency of such a practical paradigm of communication can be improved by an AE system powered by two fine-tuned LLMs. In this system, the user first types the sentence initials which get expanded to five candidate sentences by the initials AE model. If the intended sentence is not found, the system enables two revision methods: incremental spelling of words (KeywordAE) and selecting from alternative words (FillMask). After confirming the

usability of the system with non-AAC touch-typing users on a mobile device, we obtained test results from AAC eye-gaze typers with ALS showing that such users could indeed understand the UI design intentions, take advantage of the predictive power of the LLMs fine-tuned to the AE tasks, reduce the number of keystrokes entered by 14–46 absolute percentage points, and improve their typing speed by 29–60%. Greater speed gain was observed in the AAC eye-gaze users than in the non-AAC users, which was attributed to the greater benefit of motor savings in the former group of users than the latter under a trade-off relation with increased cognitive load of using the LLM-powered UI. The work shows that fine-tuned LLMs integrated with UI design could bring real benefits to people with severe motor and speech impairments.

## Methods

This study was approved by Google's AI Principles Review Committee and the Ethics Review Committee of Team Gleason. In addition, this study was conducted consistent with user research studies for Google's consumer products (https://userresearch.google.com/) and accessibility trusted tester programme. All study participants, including the able-bodied ones in the non-AAC touch-typing lab study and the two eye-gaze users with ALS, gave informed written consent prior to participating in this study.

### Large language models

We used the 64-billion-parameter version of the LaMDA[25], a pre-trained, decoder-only (LLM) as the base model for fine-tuning on the KeywordAE and FillMask tasks in SpeakFaster. LaMDA was implemented in the Lingvo[53] framework which was in turn based on TensorFlow[54]. The base LaMDA consisted of 32 transformer[55] layers (embedding and model dimensions: 8192, feed-forward dimensions: 65,536, number of attention heads per layer: 128; activation function: ReLU) and was pre-trained for the next-token prediction objective on a large-scale text corpus containing 1.56 trillion tokens from a mixture of public web text and public dialogues[25]. LaMDA used a SentencePiece tokenizer[56] with a vocabulary size of 32,000. To optimise serving efficiency (see Supplementary Section S3), we fine-tuned two models for the KeywordAE and FillMask tasks separately.

The KeywordAE models were fine-tuned on data synthesised from four publicly available dialogue datasets. These were the same dialogue datasets as used in ref. 33. Each dialogue in the datasets consisted of a number of turns, yielding multiple examples for LLM fine-tuning and evaluation. Given a turn of a dialogue, we synthesised an example with three parts: Context, Shorthand, and the Full phrase. The context consisted of the previous turns of the dialogue, separated by the curly braces as delimiters. All turns of a dialogue except the first turn were associated with a non-empty context. The shorthand was an abbreviated form of the full text generated from SpeakFaster's abbreviation rules. The three types of abbreviations that we included in the LLM fine-tuning are listed below and illustrated with the examples in Figure 11.

- Initials-only abbreviation: Abbreviating each word as the initial letter in a case-insensitive fashion (e.g. 'I saw him playing in the bedroom' ⟶ 'ishpitb'). Sentence-final punctuation was omitted, while sentence-middle punctuation (most commonly the comma) was preserved in the abbreviation (e.g. 'ok, sounds good' ⟶ 'o,sd'). These aspects of the abbreviation scheme were identical to the previous study[33]. However, to make the abbreviation rules easier to learn and memorise for users, the current study simplified the rule regarding contracted words, wherein a contracted word was abbreviated as only its initial (e.g. 'you're' ⟶ 'y' instead of 'yr' as in the previous work[33]).
- Abbreviations with complete keywords: These abbreviations contained one or more words completely spelled out, embedded in the initials-only abbreviation for the remaining words while

respecting the original word order. For example, 'I saw him playing in the bedroom' may be abbreviated as 'ishpit bedroom', where the last word 'bedroom' is spelled out. The same phrase may also be abbreviated as 'i saw hpit bedroom', where both the 2nd and last words are spelled out. They enabled users to find phrases that couldn't be expanded correctly based on the initials alone, which tended to be longer and less predictable phrases.
- Abbreviations with incomplete keywords: Similar to abbreviations with complete keywords, but allowing the keywords to be partially written. Compared to complete keywords, incomplete keywords afforded additional keystroke saving by further exploiting the redundancy and compressibility of written text. The fine-tuning of LaMDA incorporated two forms of incomplete keywords: prefix and consonant. In the prefix abbreviation scheme, a word can be abbreviated as a sequence of two or more characters that is a prefix of the word. For example, the word 'bedroom' may be abbreviated as 'be', leading to the phrase abbreviation 'ishpit be'. The consonant scheme omitted vowel letters ('a', 'e', 'i', 'o' and 'u') if they were not at the beginning of a word and kept only the first few non-vowel letters, leading to a phrase abbreviation such as 'ishpit bd' where 'bd' is the shorthand for the word 'bedroom'. The SpeakFaster UI tested in the user studies, however, only supported the prefix scheme for the sake of simplicity.

Fine-tuning of the AE model was performed on 32 Google Tensor Processing Unit (TPU) v3 chips connected in a $4 \times 8$ topology[57]. A maximum input length of 2048 tokens was used. The training used a per-host batch size of 16 and a shared AdaFactor optimiser[58] under a fixed learning rate of $2 \times 10^{-5}$. The checkpoint for subsequent serving and evaluation was selected on the basis of minimal cross-entropy loss on the dev set, which occurred at fine-tuning step 12,600.

To fine-tune LaMDA on the FillMask task, we use the same four public dialogue datasets as above. However, instead of including abbreviated forms of the phrases, we synthesised examples of a phrase with a single word masked, followed by the ground truth word. Therefore, instead of triplets of context-shorthand-full in the AE fine-tuning, the FillMask fine-tuning was based on context-phrase-word triplets such as (see also bottom right of Figure 11):

Context: {Been sitting all day. Work was just one meeting after another.} Phrase: {Oh, I'm s_.} Word: {sorry}.

In the example above, 'sorry' is the masked word. The 'Phrase' part contains the masked word with its initial letter embedded in the context words in the correct order. These examples for FillMask avoided masking words that start with a number or a punctuation mark. Based on evaluation on the Dev split, we determined the optimal sampling temperature of the FillMask to be 2.0, which was higher than the optimal sampling temperature for KeywordAE (1.0).

Supplementary Section 1 contains detailed summary statistics of the datasets used to fine-tune these LLMs and detailed results from the offline evaluation of the fine-tuned LLMs.

### User study design and details

All user study sessions were conducted in the period between late 2021 and late 2022. Each of the non-AAC user-study participants received a small compensation in the form of a gift card for their participation. Our recruitment of the non-AAC users aimed for and achieved an approximately balanced female-male gender distribution (10F, 9M). The two AAC participants recruited for this study were an opportunistic cohort. The inclusion criteria include: (1) experienced eye-gaze AAC users who use eye-tracking AAC devices to communicate in their daily lives, and (2) have cognitive abilities within normal limits.

The non-AAC touch-typing user study consisted of 19 participants. Ten participants were recruited from employees of Google working at its Cambridge, MA office; the remaining nine participants were

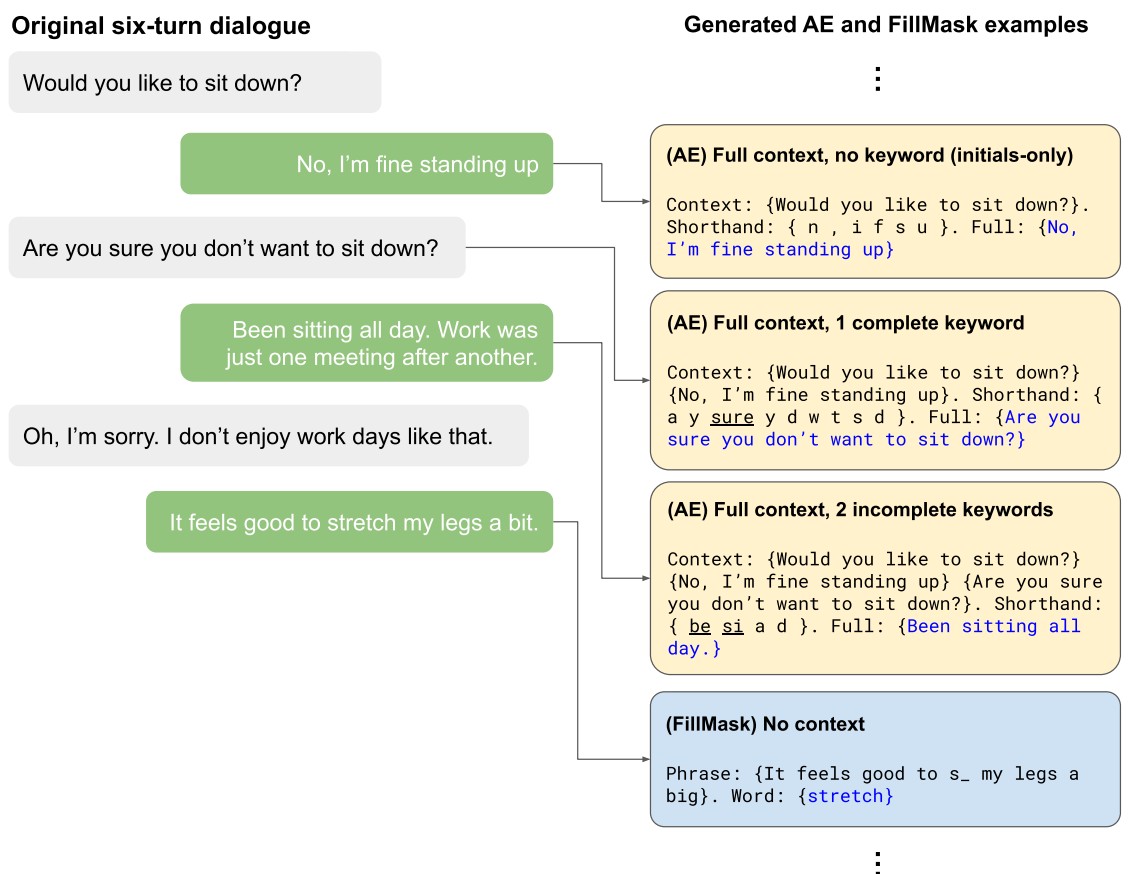

**Fig. 11 | Abbreviation schemes by examples.** On the left, we show a sample dialogue from the Turk Dialogues Corrected (TDC) dataset, which consists of six turns between two interlocutors. On the right, we show instances of individual examples generated from the dialogue, used for training the KeywordAE and FillMask models. The three AE examples vary in their abbreviation scheme. The first one consists of an initials-only abbreviation, i.e. without any spelled keywords. The second one consists of a complete (fully-spelled) keyword. The third one contains two incomplete (partially-spelled) keywords. The 'Shorthand' parts of the training examples contain the space character inserted between the letters and keywords that comprise the abbreviation, a trick employed to improve LaMDA fine-tuning accuracy by extracting more consistent tokenization from the underlying SentencePiece tokenizer[56]. The underscores for the complete and incomplete keywords are added only in this figure for visual clarity. The FillMask example at the bottom shows an instance of no-context example. The actual dataset contains both no-context examples and examples with previous turns of the dialogues as context. In these examples shown, the text in black is the inputs to the LLMs, while the text in blue is the targets that the LLMs are trained to generate. The colours are added only for visual clarity of presentation in this figure.

recruited from the Boston area. They were all adults with associate-level education or higher and were speakers of American English. During a user-study session, the user was comfortably seated, facing a Samsung Galaxy Tab S6 10.4-inch tablet placed in a stable level position on a tabletop. The device remained in landscape orientation. It presented the SpeakFaster UI running as an app at the top half of the screen and Gboard at the bottom half. Both SpeakFaster and Gboard were displayed along the full width of the tablet screen and operated by the participants through taps on the touchscreen.

In the lab studies (including the non-AAC user study sessions and the session with the AAC eye-gaze lab study participant), the ten scripted dialogues were arranged in two blocks. During the first block of five dialogues, the user played the role of the second interlocutor and entered the text for the 2nd, 4th, and 6th turns. In the second block of five dialogues, the user played the role of the first interlocutor and entered the text for the 1st, 3rd, and 5th turns. The ordering of the ten dialogues used in the first two blocks was randomised on a user-by-user basis. Following the scripted phase was an unscripted phase to measure the performance in a spontaneous text entry task in which the user typed their intended phrases instead of prescribed ones.

During the unscripted dialogues of the lab study, the experimenter operated a companion web application on a laptop to initiate test dialogues. A scripted dialogue proceeds without experimenter intervention once started. An unscripted dialogue required the experimenter to start the dialogue with a predefined question in the 1st turn, while supplying responses in the 3rd and 5th turns. The experimenter used the same companion application to send those dialogue turns. Supplementary Section 4 contains the list of the dialogues used in the scripted blocks of the lab study and the opening questions used in the unscripted block.

The participants of the AAC eye-gaze study were both experienced users of Tobii eye-trackers and their associated AAC software. The lab study participant LP1 was diagnosed with sporadic ALS 17 years prior to the study and had 13 years of prior experience using eye-gaze AAC systems. The lab study was conducted on the Tobii (R) I-16 Speech Generating Device with a built-in eye-tracker and the TD Control software that LP1 used on a daily basis. The field study participant FP1 was diagnosed with sporadic ALS 11 years prior to this study and had 10 years of experience with AAC and 10 years of experience with Tobii eye-gaze AAC products. FP1 used a Tobii Dynavox PCEye (R) Mini IS4 eye tracker with Tobii Dynavox Windows Control to operate a Microsoft Surface (R) tablet. Both participants were males in the age range of 40–50 at the time of this study.

For lab study with participant LP1, the user's word prediction history of the Tobii Windows Control virtual keyboard was cleared immediately before the start of the experiment and restored afterwards, in order to prevent personal usage history of the soft keyboard from interfering with the testing results. We note that the Tobii prediction was reset only for the lab study user and not the field study user. We additionally note that, since we do not reset the keyboard for the field study, the baseline AAC system has a slight advantage because their n-gram system would have already been adapted to the user preferences and vocabulary over an extended period of usage. Note that the KeywordAE and FillMask LLMs used in this study did not store previously-entered text or provide personalisation or adaptation to the user's own vocabulary or input history, despite their utilisation of the previous turns of the current dialogue as a contextual signal to enhance their accuracies for all non-AAC participants and AAC participants.

### Reporting summary

Further information on research design is available in the Nature Portfolio Reporting Summary linked to this article.

## Data availability

Data used to train and evaluate the LLMs in this paper are available from existing public sources. We list the training data sources and code for their processing in the data directory of the TeamGleason SpeakFaster GitHub repository: https://github.com/TeamGleason/SpeakFaster/tree/main/data/naacl_2022_suppl_data. The stimuli used in the user study are available in Supplementary Information. The raw data from the user studies are not publicly available due to data privacy laws. The anonymized data are available under restricted access for user privacy. Access can be obtained by sending a request to the corresponding author for academic purposes only. The processed and summarised forms of the user-study data that are presented in the figures and table of this article are available for access at: https://github.com/TeamGleason/SpeakFaster/tree/main/data/user_study_paper_data. Source data are provided with this paper.

## Code availability

The code for training and evaluating the LLMs is based on Google's proprietary LaMDA system[25] and is not made available publicly. The code for the user interface and the scripts for data analysis are made available in the TeamGleason SpeakFaster GitHub repository: https://github.com/TeamGleason/SpeakFaster/tree/main.

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

## Acknowledgements
Mahima Pushkarna assisted with the design of the SpeakFaster UI. Jon Campbell assisted with Windows programming and eye-tracker interfacing. William Ito and Anton Kast helped with UI prototyping and testing. Julie Cattiau, Pan-Pan Jiang, Rus Heywood, Richard Cave, James Stout, Jay Beavers and John Costello provided insightful discussions and other assistance. Shumin Zhai provided extensive feedback on the manuscript. We are grateful to Tobii (R) for granting us permissions to use the Tobii Stream Engine for eye-gaze prototype development.

## Author contributions
S.C. and S.V. verified the data in the study and took responsibility for the integrity of the data and the accuracy of the data analysis. S.C., S.V., S.M.G., P.Q.N. and M.P.B. conceived and designed the study. S.C., S.V., K.T. and X.X. trained and evaluated the large language models. S.J. and S.C. designed the user interface. S.C., K.S., R.L.M., D.V., E.K. and B.C. were involved in data collection. S.C., S.V., K.S. and X.X. analysed the data. A.N., S.K., M.R.M., P.Q.N. and M.P.B. provided strategic guidance and oversight. S.C., S.V., X.X. and S.K. drafted the manuscript with input from all authors. The final version of the article has been approved by all the authors.

## Competing interests
S.C., S.V., X.X., K.T., M.R.M., S.K., A.N., R.L.M., P.Q.N. and M.P.B. are employees of Google and own stocks in the company. E.K., D.V., B.C. and S.M.G. are members of Team Gleason. We declare no other competing interests.
