## [Peer Review File · Nature Communications]

Reviewers' Comments:

Reviewer #1:

Remarks to the Author:

The paper applies fine-tuning of language models to augmentative and alternative communication (AAC). The strong use case and user study sections add significant value to this paper. Additionally, the study design was solid, and appropriate statistical values were presented. The paper is well-written, easy to understand.

While technical details for implementations especially the fine-tuning steps are absent from the main manuscript, they are available in the supplementary section. Several interesting sections are located in the appendix. The GitHub repository is open-source and accessible online. I have provided a suggestion regarding the repository in a separate section of the questionnaire.

Overall, this is a strong paper with potential for high impact on both the research community and society.

I have a few questions to pose:

To implement LLM-based methods for real-world use, lowering latency is critical. In line with this, the authors covered this topic in Section 5.4 entitled "Latency and internet access" and Section A.3 in the Supplementary Information. In addition to currently introduced serving methods, have you considered other optimisations, such as model distillation, or different quantization precisions, such as Int8?

Distillation and quantization may enable the model to run on mobile applications, depending on the final model size.

Additionally, was there a decrease in performance caused by application of quantization?

In the discussion section (Section 5.3), the authors mentioned prompting. However, I believe this task is highly specialized and not likely to be learned during pre-training from general texts.

I wonder how LLMs will perform in zero-shot and few-shot experiments. Out of curiosity, have you seen any results from prompt-tuning experiments?

Minor points:

On page 3, Figure 1 is crucial for understanding Section 2. However, the figure is indistinguishably small, particularly in print.

On page 25, the term "tokens4" in the last paragraph seems odd to me. Could it be a

typo?

Overall figures in this paper are not easy to read. Figure 1 and 2 are too small to read when printed and labels in Panel B of Figure 7 is pretty hard to read.

Reviewer #2:

Remarks to the Author:

Based on the abstract for this review I was looking forward to reading this paper and overall I was not disappointed. This is relevant, novel, significant work that indicates future impact on the AAC field. As an AAC practitioner-researcher however there were a few elements which I feel need addressing prior to publication, a couple of which (the first two below) I would class as significant. Hopefully the authors agree that addressing these points will improve the impact and clarity of the study, and hopefully also make it more likely to have impact in the AAC field (a problem in this space I think being that this type of paper tends to be read only by HCI researchers are vice-versa).

1. Study recruitment methodology and ethical review. This study included human participants and as such should have had ethical review and met appropriate standards RE informed consent etc. I assume it has, but the details of this need including. In addition further details on the identification and recruitment methods used to recruit participants should be included, and the relevant limitations in this method discussed (in relation to the applicability of generalisation of the results) in the limitations section. Further details of the participants and their AAC systems should be included if at all possible (suggestions in PDF).

The limitations should also discuss any likely biases that existed in the participants in terms of the wider population of those living with ALS.

2. It took me to the end of the paper to realise that the SpeakFaster system was including the conversational turns of the communication partner in the prediction context. Firstly, this should be made much clearer earlier in the paper as for me (looking at the implications for AAC practice) this significantly impacts my interpretation of the results. Secondly, the limitations of this in terms of the ecological validity of this as an AAC system (as compared to a text based communication system) should be expanded on.

I would have appreciated this study even more if the experimental comparisons had been between 'normal AAC method', 'AAC method including LLM context of other's turn' and 'SpeakFaster abbreviation expansion including context of other's turn.

Reading A4 I also note that the Tobii prediction was reset prior to the tests - this needs to be clearer in the main text (and was it completed for both participants?) - I also think this needs discussing in the limitations - as it is not comparing with the 'best available alternative' - and ? actually puts the SpeakFaster system at a learning advantage ? (I'm not clear on this as I can't unpick what use of the Tobii system happens between it being reset and the test conditions)?

Further, the description of the 'field trial' needs expanding, on first reading I assumed this was using more naturalistic communication, but actually i think it is the same experiment (silent/typed conversation) with the only difference being it being in the participants' own setting?

3. The terms used for the two groups of participants in the study need reviewing and making consistent throughout. I do not think 'mobile group' and 'eye gaze group' is appropriate as this implies the only difference is in access method. Similarly I think the title should be reviewed, so as to reduce/remove any implication that these results may be generalizable across all those with motor impairments. e.g. "Using Large Language Models to Accelerate Communication for eye gaze typing users with ALS" is more accurate.

4. I think the paper would benefit from a (little) bit more context and introduction to the AAC literature (A few specific points in the background section in the attached PDF).

I have made some other more minor points in the attached PDF (comments are not finessed, please take these constructively, as they are meant as such!) - mainly just suggesting moderation of a couple of statements and suggestion RE heading structure and/or wording. The results are also, at one point, over-simplified as the fact that one of the results did not reach significance is missed in summary.

Thanks again for the paper and taking forward this work.

SpeakFaster Nature Comm. submission

Point-by-point response to reviews

We thank the reviewers for their time and valuable feedback. We have copied the original reviewer comments (in **black**) and added our responses (in **purple**) in this document. We have also attached an updated PDF with the additions marked in blue, and the format changed as per Nature Communications formatting guidelines.

REVIEWER COMMENTS

Reviewer #1 (Remarks to the Author):

The paper applies fine-tuning of language models to augmentative and alternative communication (AAC). The strong use case and user study sections add significant value to this paper. Additionally, the study design was solid, and appropriate statistical values were presented. The paper is well-written, easy to understand.

While technical details for implementations especially the fine-tuning steps are absent from the main manuscript, they are available in the supplementary section. Several interesting sections are located in the appendix. The GitHub repository is open-source and accessible online. I have provided a suggestion regarding the repository in a separate section of the questionnaire.

Overall, this is a strong paper with potential for high impact on both the research community and society.

We thank the reviewer for the constructive feedback and for appreciating our motivation, the user study design, and the evaluations. We also appreciate that the reviewer took time to checkout the codebase and provided suggestions for improvements. We have addressed your comments below and in the attached revised pdf.

I have a few questions to pose:

To implement LLM-based methods for real-world use, lowering latency is critical. In line with this, the authors covered this topic in Section 5.4 entitled "Latency and internet access" and Section A.3 in the Supplementary Information. In addition to currently introduced serving methods, have you considered other optimisations, such as model distillation, or different quantization precisions, such as Int8?

Distillation and quantization may enable the model to run on mobile applications, depending on the final model size.

The reviewer is correct in pointing out the potentials for quantization and distillation for reducing the size and inference latency of the model. Conversion to a lower-precision data type (to the bfloat16 data type) is already performed in our system (See Appendix A.3). We did not perform model distillation - we leave that for future studies. Below we'll address some points regarding performance and user experience and then share more on what might be possible in future works.

[Optimization and model size] While we have not worked on size optimization of the tuned model, we have studied the effect of model size in our previous work [Cai et. al. NAACL 2022]. Specifically, smaller models, though significantly faster, also have much lower prediction accuracy than larger models, which likely will lead to significantly worse user experience compared to the 64B model we used in the user study. Here are evaluation plots from the previous work to show that

- (1) The 4B parameter model has much lower latency, but
- (2) The accuracy (red dotted line) of the 4B model is far worse (~65% absolute) compared to the large model (which achieves ~80% absolute accuracy)

All these models were tuned and evaluated on the same dataset.

[On-device models] As mentioned above, the idea of performing distillation and quantization to get a small model, presumably to be integrated on-device, is an idea that should be valuable to explore for the AAC community. However, a practical issue is that even the smallest on-device large language models (e.g. 2B-2.5B param models, e.g., Gemini Nano-1 [1]) not only lack in performance but also tend to consume a lot of resources and will likely drain the battery and CPU cycles of the device. The power consumption may become a limiting factor due to the frequent invocation of the model (after every keystroke) and the fact that the eye gaze AAC system of many users rely on a battery with limited capacity attached to their wheelchairs.

That being said, the open source community is actively researching on-device smaller LLMs, so an interesting future direction would be to train such a model on the data we used for training. We note that all the datasets used to train our models are publicly available! While we are not allowed to share them directly due to licensing restrictions, we have now added a [README file in GitHub](https://github.com/TeamGleason/SpeakFaster/tree/main/data/naacl_2022_suppl_data) pointing to the data (https://github.com/TeamGleason/SpeakFaster/tree/main/data/naacl_2022_suppl_data)

Additionally, was there a decrease in performance caused by application of quantization?

As mentioned above and in the manuscript, our fine-tuned LaMDA model is converted to the lower-precision and more computationally-efficient bfloat16 data type from the original float32 data type during training. This quantization led to minimal decrease in the abbreviation-expansion accuracy (<1 percentage point).

In the discussion section (Section 5.3), the authors mentioned prompting. However, I believe this task is highly specialized and not likely to be learned during pre-training from general texts. I wonder how LLMs will perform in zero-shot and few-shot experiments. Out of curiosity, have you seen any results from prompt-tuning experiments?

In the second plot above, the solid orange line (BaseLLM*) shows the performance of the best few-shot experimental results we observed on the largest pre-trained LLM. As you have rightly noted, since the task is highly specialized, zero-shot and few-shot accuracy are much lower compared to fine-tuned models and hence not suitable for practical use by AAC users.

Minor points:

On page 3, Figure 1 is crucial for understanding Section 2. However, the figure is indistinguishably small, particularly in print.

We have now split Figure 1 into three separate figures to make the screenshots and flowcharts in them easier to read.

On page 25, the term "tokens4" in the last paragraph seems odd to me. Could it be a typo?

The "tokens4" was a typo. It is fixed now.

Overall figures in this paper are not easy to read. Figure 1 and 2 are too small to read when printed and labels in Panel B of Figure 7 is pretty hard to read.

In addition to the aforementioned split of Figure 1, we have also split Figure 2 into two figures (one for the simulation strategies and one for the simulation results). For Figure 7 (new Figure 10), we have reworked the legend of the time-interval breakdown plot in Panel B to make it more readable.

Reviewer #1 (Remarks on code availability):

The repository is well-documented but requires minor improvements for code reproduction. The README is easy to follow, but some information is missing.

Note: I have been familiar with python (machine learning projects) and open-source projects for over 10 years. I barely have experience with Node, Typescript, or C#.

Code reproduction comments:

* The repository mentions 'Node 14+' and provides a link to the official Node website. It would be better to specify the exact version of Node, as I had to spend a few minutes figuring out how to run it with the version from the Node website (it was v20). After a web search and adding the option `NODE_OPTIONS=--openssl-legacy-provider`, I was able to run it. According to web sources, users may face these errors starting from version 17.

Thanks for checking out our code and for this helpful suggestion. We have edited the github repo with the updated Node version and information about the openssl flag.

Reviewer #2 (Remarks to the Author):

PDF with the reviewers comments:  2_reviewer_attachment_1_1706200398_convrt.pdf

Based on the abstract for this review I was looking forward to reading this paper and overall I was not disappointed. This is relevant, novel, significant work that indicates future impact on the AAC field. As an AAC practitioner-researcher however there were a few elements which I feel need addressing prior to publication, a couple of which (the first two below) I would class as significant. Hopefully the authors agree that addressing these points will improve the impact and clarity of the study, and hopefully also make it more likely to have impact in the AAC field (a problem in this space I think being that this type of paper tends to be read only by HCI researchers are vice-versa).

We thank the reviewer for your thoughtful review and suggestions, which helped us improve the overall quality of this manuscript. We have incorporated many of your suggestions, and have revised the paper and provided clarifications and responded to your comments below.

1. Study recruitment methodology and ethical review. This study included human participants and as such should have had ethical review and met appropriate standards RE informed consent etc. I assume it has, but the details of this need including. In addition further details on the identification and recruitment methods used to recruit participants should be included, and the relevant limitations in this method discussed (in relation to the applicability of generalisation of the results) in the limitations section.

Further details of the participants and their AAC systems should be included if at all possible (suggestions in PDF).

The limitations should also discuss any likely biases that existed in the participants in terms of the wider population of those living with ALS.

This study was conducted consistent with user research studies for Google's consumer products (<https://userresearch.google.com/>) and accessibility trusted tester program (e.g. https://docs.google.com/forms/d/e/1FAIpQLSfcb-IOmCZ__09SSyFAul_k2WBLR05URYbR_Stv9N42u7GTiw/viewform). Additionally, we sought and received approval by the ethical review committees of Google and Team Gleason. All study participants, including the able-bodied ones in the non-AAC touch-typing lab study and the two eye gaze users with ALS, gave informed consent prior to participating in this study.

The participants recruited for this study were an opportunistic cohort. The inclusion criteria include: 1) experienced eye-gaze AAC users who use eye-tracking AAC devices to communicate in their daily lives, and 2) have cognitive abilities within normal limits.

This information is added to the supplementary materials of the manuscript.

2. It took me to the end of the paper to realise that the SpeakFaster system was including the conversational turns of the communication partner in the prediction context. Firstly, this should be made much clearer earlier in the paper as for me (looking at the implications for AAC practice) this significantly impacts my interpretation of the results. Secondly, the limitations of this in terms of the ecological validity of this as an AAC system (as compared to a text based communication system) should be expanded on.

Based on this feedback, we have strengthened the emphasis on the utilization of conversational context in early parts of the manuscript, including the abstract, Introduction, and Approach. In addition, we added more details to the part of the Discussion dedicated to the ecological considerations surrounding context.

I would have appreciated this study even more if the experimental comparisons had been between 'normal AAC method', 'AAC method including LLM context of other's turn' and 'SpeakFaster abbreviation expansion including context of other's turn'.

This is actually consistent to how we do the study, and we have tried to edit the manuscript to incorporate your suggested phrasing. Specifically,

- (1) The "Baseline" in Figure 6 and Figure 7 in Sections 2.5 and 2.6 represents the "normal AAC method" where the eye-gaze ALS participants use the smart keyboard completion available on their eye-gaze enabled device.
- (2) The simulated "Gboard baseline" (blue) in Figure 5 (Panels A and B) do represent the "Normal AAC method including LLM context of other's turn". Note that the normal AAC method does not fully take in other's turns as context, but in the simulation evaluations we are able to take the context into account.
- (3) Figure 5 (Panels A-C) also provides a comparison of the SpeakFaster abbreviation expansion strategies *without including context* (gray) and SpeakFaster abbreviation expansion strategies *with full context* (yellow).

Figure 5. Panel A

Figure 5. Panel B

Reading A4 I also note that the Tobii prediction was reset prior to the tests - this needs to be clearer in the main text (and was it completed for both participants?) - I also think this needs discussing in the limitations - as it is not comparing with the 'best available alternative' - and ? actually puts the SpeakFaster system at a learning advantage ? (I'm not clear on this as I can't unpick what use of the Tobii system happens between it being reset and the test conditions)?

The Tobii prediction was reset only for the lab study user and not the field study user.

We should note here that actually for the Field Study the advantage is on the AAC system side because their n-gram system would have already been adapted to the user preferences and vocabulary over extended months/years of usage.

For the lab study there should be no difference or advantage one way or another. Note that LLM is stateless and doesn't have personalization. But for conversation turns the user chooses to provide the context.

We have now made an edit to include this in supplement A4.

Further, the description of the 'field trial' needs expanding, on first reading I assumed this was using more naturalistic communication, but actually I think it is the same experiment (silent/typed conversation) with the only difference being it being in the participants' own setting?

Your understanding is correct. It is silent typed conversations. The participant is unable to speak and the participant is in their natural environment.

3. The terms used for the two groups of participants in the study need reviewing and making consistent throughout. I do not think 'mobile group' and 'eye gaze group' is appropriate as this implies the only difference is in access method. Similarly I think the title should be reviewed, so as to reduce/remove any implication that these results may be generalizable across all those with motor impairments. e.g. "Using Large Language Models to Accelerate

Communication for eye gaze typing users with ALS" is more accurate.

This is a great point. We have now clarified this throughout the paper. We use the terms AAC eye-gaze users (or AAC users) and non-AAC touch-typing users (or non-AAC users) to refer to the two groups of users. We have also edited the title as suggested.

4. I think the paper would benefit from a (little) bit more context and introduction to the AAC literature (A few specific points in the background section in the attached PDF).

In addition to citing the additional previous studies suggested by the reviewer, we have added some sentences and references to the Introduction to describe the context of AAC use and challenges experienced by AAC users.

Reviewer #2 (Remarks on code availability):

I have checked the code is available on Github but not compiled/reviewed it.

Thank you for your remarks!

Reviewer #2 comments in the PDF

- Title: changed "Users with Severe Motor Impairments" to "Eye Gaze Typing Users with ALS"

Introduction

- Changed citation to AAC in intro

- Add better reference to support 190WPM speaking rate
 - 125-185 WPM Swiffin et al Adaptive and predictive techniques in a communication prosthesis
 - We have now added a citation to this previous paper in the Introduction section.
- Useful to add in information about AAC speaking rates (e.g. in other cohorts).
 - Telling tales: unlocking the potential of AAC technologies - Waller - 2019 - International Journal of Language & Communication Disorders This paper has some good references under “Communication Rates”
 - We are now citing this paper in the Introduction section.
- There are many other access methods to AAC than BCI and eyegaze, and other ways of rate enhancement. I appreciate there is more active research interest in BCI, but this should be presented in context of the other existing methods and techniques.
 - Using NLG to Help Language-Impaired Users Tell Stories and Participate in Social Dialogues
 - a cross-cultural study of picture-based sentences constructed by English and Japanese speakers: Augmentative and Alternative CommunicationA Design Engineering Approach for Quantitatively Exploring Context-Aware Sentence Retrieval for Nonspeaking Individuals with Motor Disabilities | Proceedings of the 2020 CHI Conference on Human Factors in Computing Systems
 - We thank the reviewer in pointing out these additional relevant previous studies. We are now citing them in the appropriate place in the Introduction and Discussion sections of this manuscript.
- can see that the argument is that abbreviation expansion likely reduces keystrokes and thus reduces motor demands - however this is a separate point (and should be made separately) - and saying for 'each keystroke' is then wrong?
 - Rephrased to say: major bottleneck to faster gaze typing for disabled users is the eye fatigue and temporal cost associated with performing many keystrokes.
- Grid3 uses swiftkey.
 - Added a citation.
- spans of correct predictions are short and the utilization of the predictions maybe too infrequent to offset costs ...
 - This is a more contested point than presented here - I think it is accepted that this depends on a number of user and system factors. I think 'may be too infrequent' is more appropriate.
 - E.g <https://dl.acm.org/doi/10.5555/1722763.1722768>
 - Rephrased and added a citation
- Add citations for “highly abbreviated English text inputs”
 - Suggestion was “Waller etc.”
 - We added a citation to Gorman et. al. and also referenced Demasco et. al.

Methods

- “conversational context” ⇒ other speaker’s turn
 - Done revising this.

We would like to point out that “conversational context” includes two parts: the user’s own previous turns and the other speaker’s (i.e., conversation partner’s) turns. The system tested in this manuscript utilized both. This is clarified in the manuscript.

Simulation Results

- Figure 2 caption - yes orange bars include conversational context which encompasses the other speaker’s turn and also potentially conversation history from this specific multi-turn dialog.
 - We have added this clarification to the caption of the figure in question (new number: Figure 5).

User studies

- Clarify that user studies are entirely text-only conversations
We have clarified this in the User Study section, followed up with additional related discussions in the Discussion section.
- Having got to the end, I understand that this was an entirety text-only conversation? This needs to be clearer.
- Having read A4 and studied Figure 1 better, I now see that the LLM is including the context of the question / partner's utterances?
 - 1) this needs to be clear in the main text

We have beefed up the emphasis on the utilization of the conversational context throughout the manuscript, including a more in-depth discussion of this topic in the Discussion section.

2) this is a significant challenge to the ecological validity of the study, as this is not a normal AAC situation.

3) this is a challenge to the comparability with this and the standard AAC keyboards ... It leads me to ask the question would not a better comparison be against a standard keyboard/word prediction layout but with the prediction based on an LLM which included this partner conversation as context?

Our comparisons include this. And our response to your comments earlier between 'normal AAC method', 'AAC method including LLM context of other's turn' and 'SpeakFaster abbreviation expansion including context of other's turn' with references to Figure 5 addresses this.

- Tobii eye gaze keyboard – expecting more information later

Added more info about the eye tracker and AAC software setups for both the lab and field study participants.

- Change “mobile users” term – not the term used for the group above (non disabled users). Turning this into 'mobile users' slightly unintentionally implies that disabled people wouldn't use mobiles.
- Speakfaster enables significant keystrokes savings [added: in the non-AAC user group]
- latencies of calls to cloud-based LLMs.
 - Answered in the comments above

- “likely” reflects

These four suggestions were incorporated into the revised manuscript.

Eye Gaze user studies

- Add more information on recruitment of eyegaze users
 - addressed in the main comments.
- edited cognitive abilities were reported within normal limits.
- On Tobii eye gaze keyboard set up
 - this should be more specific, which app / what platform was the device on?
 - + what selection method (dwell? what period?)
 - More information on the participants also required to allow these data to be better interpreted. E.g. age, type of ALS, prior experience with, or comfort with, IT, length of time using AAC and then AAC with eye-gaze
 - Added info about the two AAC study participants, including type of ALS, years of living with the diseases, durations of using AAC and eye-trackers prior to this study, and the hardware and software setup of the eye-gaze AAC systems.
- (70.7%), indicating that the user occasionally failed to choose the correct words when they appeared.
 - #I find this impact of 'errors' interesting, and it reflects clinical experience and other AT literature, and I hope this is discussed later...

This is included in the discussions under “cognitive overhead” and also in the supplement A.5
- [Clarification] success rate of the initial AE and FillMask was lower for unscripted dialogues reflects the domain mismatch between the user’s personal vocabulary and the model’s training corpus
 - Comment] should this be “likely” reflects

This is known to be true. The scripted dialogues are unseen test conversations about everyday life that are similar in form to some dialogs in the training corpora.
- incorrect reference to SpeakFaster Observer that reports FP1’s typing rate.
 - We have corrected this.
- Field Study FP1 engaged in test dialogue with the experimenter
 - I dont understand how this is more natural use? When i first read this I thought this was a field trial as in used in natural settings and communication situations. In fact, I think the experimenter is still typing in the communication partner's turns and the setting is still ?scripted or non scrypted? i.e. it is the same experiments just in a different setting (what setting? home, work, cafe?)
 - Why is this section of the study not included in detail in Appendix A4?

This has now been clarified in Sec. 3.2
- Section 4.7: 11,189 ms
 - These sentences are really hard to process and would benefit from revision.
 - I don't understand why the prior measures are per word, whereas this is a total ms - total of what? per test?). I would find it easier if there were 2 sentences with identical format comparing those without ALS using the mobile app and those with ALS using SpeakFaster.

Thanks for sharing, we have edited this.

On the 11,189 ms Thanks for catching it. We edited to say ms/word.

- a significant advantage of the SpeakFaster UI of text input compared to the baseline of conventional eye-typing systems with forward-prediction.
 - This is over-simplifying, as the results did not all reach significance - i.e. the unscripted lab results for FP1 were not significant.

Edited to say considerable advantage since there was a rate enhancement in all cases.

Discussion

- “builds on user knowledge”
 - I doubt the AAC users in this study had previously used abbreviation expansion.
 - My impression is that it has fallen out of use in both AAC and text messaging (not abbreviations, but abbreviation expansion)? This should be stated above in participant characteristics if they have.

It is true that the AAC users did not have prior experience using abbreviation expansion. Our comparison was with regard to us (as developers as SpeakFaster), explaining the specific first-letter based abbreviation expansion scheme to the participants. The knowledge and familiarity with SMS-style texting made it easy for participants to grasp the idea of the scheme although they didn't have prior experience with using it for open-ended phrases and full sentences.

- motor impaired users → eye gaze ALS users
 - We revised this according to the suggestion.
- Latency and access to the internet.
 - Does it imply a certain speed of access too - as in real use this (and mobile data coverage) will be a significant factor. - i.e. is the latency predominately processing (I assume so, but again, it would be good to state).

Thanks, we have edited the discussion. This is a purely text-based application and can piggyback on text messaging application bandwidth but will nevertheless require internet access via Wi-Fi or mobile data coverage as you have noted.

- Eye-gaze users with ALS → eye-gaze typing users with ALS
 - We revised this according to the suggestion.

Limitations

- Need to discuss limitations in the applicability of the user study results and recruitment method - i.e. that the participants are likely not representative of most of those living with ALS.

Thank you. We addressed this in the main comments above and included it in the limitations under Discussion.

- More clarity on conversational context
 - As mentioned above, we have increased emphasis on the use of conversational context.
- Able-bodied
 - Edited to non-AAC touch typing users
- attributable → attributed
 - Revised accordingly.

Supplementary

- A4 [clarify] silent typed conversation, that went into the context of the LLM
- For lab study with participant LP1, the user's word prediction history of the Tobii Windows Control virtual keyboard was cleared immediately before the start of the experiment, in order to prevent personal usage history of the soft keyboard from interfering with the testing results.
 - Interesting. Should be in the main document and in the limitations?
I appreciate that the LLM hasn't learnt context, but equally it is not comparing against the best available alternative/ reduces ecological validity.
 - User system was at a disadvantage
 - No. We clarified this in the main comments earlier.

References:

1. Google Gemini Team (2023) Gemini Technical Report.
https://storage.googleapis.com/deepmind-media/gemini/gemini_1_report.pdf

Reviewers' Comments:

Reviewer #1:

Remarks to the Author:

I have carefully reviewed all the responses to my previous comments and questions. The authors provided detailed explanations and responded in a reasonable manner with related article.

Reviewer #2:

Remarks to the Author:

Thanks for the complete response to the review and corresponding amendments. These have clarified my questions and concerns and I have no further comments and look forward to seeing this in print, testing the software and future replication/extension studies!